# Nearly-Linear Time and Massively Parallel Algorithms for $k$-Anonymity

**Kevin Aydin** *
Google Research
kaydin@google.com

**Honghao Lin**†
Carnegie Mellon University
honghaol@andrew.cmu.edu

**David P. Woodruff**
CMU & Google Research
dwoodruf@andrew.cmu.edu

**Peilin Zhong**
Google Research
pz2225@columbia.edu

## Abstract

$k$-anonymity is a widely-used privacy-preserving concept that ensures each record in a dataset is indistinguishable from at least $k - 1$ other records. We revisit $k$-anonymity by suppression and give an $O(k)$-approximation algorithm with a nearly-linear runtime of $\widetilde{O}(nd + n \cdot (n/k)^{1/C^2+o(1)})$ for any constant $C$, where $n$ is the number of records and $d$ is the number of attributes. Previous algorithms with provable guarantees either (1) achieve the same $O(k)$ approximation ratio but require at least $O(n^2 k)$ runtime, or (2) provide a better $O(\log k)$ approximation ratio at the cost of an impractical $O(n^{2k})$ worst-case runtime for general $d$ and $k$. Our algorithm extends to the Massively Parallel Computation (MPC) model, where it gives an MPC algorithm requiring $\widetilde{O}(\log^{1+\varepsilon} n)$ rounds and total space $O(n^{1+\gamma}(d + k))$. Empirically, we also demonstrate that our algorithmic ideas can be adapted to existing heuristic methods, leading to significant speed-ups while preserving comparable performance. On the hardness side, we study the related single-point $k$-anonymity problem, where the goal is to select $k - 1$ additional records to make a given record indistinguishable. Assuming the dense vs random conjecture in complexity theory, we show that for $n = k^c$, no algorithm can achieve a $k^{1-O(1/c)}$ approximation in $\text{poly}(n)$ time, providing evidence for the inherent hardness of the $k$-anonymity problem.

## 1 Introduction

As data becomes increasingly central to decision-making, research, and business intelligence, ensuring privacy while preserving data utility has become a critical challenge. Many datasets contain sensitive information, such as health records, financial transactions, or social behavior. However, even after removing direct identifiers (e.g., names, social security numbers, etc.), inadequate safeguards can still lead to re-identification. To mitigate these risks, privacy-preserving data publishing techniques have become essential for balancing data utility and privacy protection, with $k$-anonymity [Swe02] standing out as a foundational approach.

$k$-anonymity, introduced by [Swe02], ensures that each record in a dataset is indistinguishable from at least $k - 1$ other records based on a set of quasi-identifiers. These quasi-identifiers, such as age, ZIP code, and gender, may not be uniquely identifying on their own but can enable re-identification when combined with external data sources. By applying generalization and suppression techniques,

---

*Equal Contribution.

†Part of the work was done while Honghao Lin was a student researcher in Google Research.

39th Conference on Neural Information Processing Systems (NeurIPS 2025).

$k$-anonymity reduces the risk of re-identification while preserving data utility for analysis. An example can be found in Table 1. In this paper, we will only consider the case of suppression where each entry of every attribute is either included in the output, or replaced with the '$\star$' character.

Most research on $k$-anonymity has focused on finding the optimal (or near-optimal) $k$-anonymous dataset. That is, the one that minimizes the number of hidden attributes and thereby best preserves the original data. The work of [MW04] demonstrated that finding the optimal solution is NP-hard but provided an $O(k \log k)$-approximation algorithm with a runtime exponential in $k$. Later, [AFK+05] improved this to an $O(k)$-approximation with a runtime of $O(n^2 k)$. Subsequently, [PS07, KT12] further enhanced the approximation to $O(\log k)$, though their algorithm has a worst-case runtime of $O(n^{2k})$. In addition to algorithms with provable guarantees, other studies have proposed heuristics for various anonymization approaches. For example [LDR06] introduced a heuristic algorithm for $k$-anonymization of quasi-identifiers, utilizing a construction similar to $k$-$d$ trees, [DXTK15] by freeform generalization and [BKBL07, ZWL+18] proposed heuristics based on clustering.

| Age | Marital status | Home country | Gender | Age | Marital status | Home country | Gender |
|------|------|------|------|------|------|------|------|
| 20~29 | Single | USA | Male | 20~29 | Single | USA | $\star$ |
| 30~39 | Divorce | China | Female | 30~39 | $\star$ | $\star$ | Female |
| 20~29 | Single | USA | Female | 20~29 | Single | USA | $\star$ |
| 30~39 | Separation | Korea | Female | 30~39 | $\star$ | $\star$ | Female |

Table 1: An example of 2-anonymization [PS07]

Despite the extensive body of research on $k$-anonymity, many questions and challenges remain unresolved. First, the fastest algorithm for $k$-anonymity with a provable guarantee has a runtime of $O(n^2 k)$, where $n$ is the number of data points. As the size of data continues to grow in many scenarios, even an $O(n^2)$ runtime may become impractical. This raises the question of whether an algorithm with linear runtime in $n$ is possible.

> *Question* 1*: Is there an algorithm for $k$-anonymity that runs in linear time in $n$ while providing a provable approximation guarantee?*

Second, to the best of our knowledge, no work has studied $k$-anonymity in the context of massively parallel computation. In fact, most existing algorithms for $k$-anonymity rely on sequential processing. Therefore, there is strong motivation to develop parallel algorithms that can leverage the power of distributed computing frameworks to achieve faster and more efficient solutions.

> *Question* 2*: Is it possible to design an algorithm for $k$-anonymity in the massively parallel computation model while minimizing the number of communication rounds?*

Also, although [PS07] introduced improvements to achieve more practical runtimes for their $O(\log k)$-approximation algorithm, these improvements may only be effective when the dimension $d$ of each point is small. Moreover, their worst-case runtime still remains $O(n^{2k})$. A natural question arises:

> *Question* 3*: Is it possible to develop an algorithm with an $o(k)$ approximation ratio and a worst-case runtime polynomial in $n, d,$ and $k$?*

## 1.1 Our Contributions

We present the first $O(k)$-approximation algorithm for $k$-anonymity with a nearly-linear runtime.

**Theorem 1.1.** *Given a table $T$ with $n$ records $r_i \in \Sigma^d$ $(i = 1, 2, \ldots, n)$, there is an algorithm that runs in time $\widetilde{O}\left(nd + n \cdot (n/k)^{1/C^2 + o(1)}\right)$ and with high probability outputs an $O(k)$-approximation to the $k$-anonymity problem on $T$, where the constant hidden in the approximation ratio depends on $C$.*

For $C$ large, the running time approaches $\widetilde{O}(nd + n^{1+o(1)})$, which is nearly the time to read the input. Our algorithm can also be extended to the Massively Parallel Computation model (Section A.1) with a number of communication rounds that is logarithmic in $n$.

**Theorem 1.2.** *Given a table $T$ with $n$ records $r_i \in \Sigma^d$ $(i = 1, 2, \ldots, n)$, let $\gamma, \varepsilon \in (0, 1)$. There is a fully scalable MPC algorithm that outputs an $O\left(\frac{\log^2(1/\varepsilon)}{\gamma} \cdot k\right)$-approximation to the $k$-anonymity*

*problem on T with high probability. The algorithm takes $O\left(\frac{\log 1/\varepsilon}{\gamma} \cdot \log^{1+\varepsilon}(n) \log \log(n)\right)$ parallel time and $\widetilde{O}\left(nd + n^{1+\gamma+o(1)} \cdot k\right)$ total space.*

To obtain a better understanding of our third question, we propose and study the following single-point $k$-anonymity problem, where the goal is to select $k - 1$ additional records to make a given record indistinguishable while minimizing the number of hidden attributes among these $k$ points. Assuming the dense vs. random conjecture in complexity theory, we show that for $n = k^C$, no algorithm can achieve a $k^{1-O(1/c)}$ approximation in $\text{poly}(k)$ time.

**Theorem 1.3.** *Assume Conjecture 4.3, and let $n = k^C$ and $d \geq k$. There is no algorithm which runs in polynomial $n$ time that with high probability can output a $k^{1-O(1/C)}$-approximation to the single-point $k$-anonymity problem, even if each record is binary.*

Theorem 1.3 provides evidence of the inherent hardness of the $k$-anonymity problem. We remark that an open question here is whether we can extend this lower bound to the original $k$-anonymity problem and obtain a similar hardness.

## 1.2    Related Work

In addition to the work mentioned above, several studies have explored special cases of $k$-anonymity. For example, [AFK+05] proposed a 1.5-approximation algorithm for 2-anonymity and a 2-approximation algorithm for 3-anonymity. Similarly, [BDVDP13] presented a polynomial-time algorithm for the case when both $d$ and $|\Sigma|$ are constant.

On the hardness side, [BDVDP13] showed that finding the optimal solution is $w[1]$-hard with respect to the value of the solution (and $k$). Furthermore, [BDVD11] demonstrated that $c$-approximation is hard for a fixed constant $c$ in the following cases: (1) $|\Sigma| = 2$ and $k = 3$, or (2) $d \geq 8$ and $k = 4$.

Several studies have extended the definition of $k$-anonymity by introducing additional constraints, such as $l$-diversity [MKGV07] and $t$-closeness [LLV06], to enhance privacy protection for non-quasi-identifiers. Additionally, [CFL10] introduced the concept of $k$-isomorphism in social network graphs, which means that the graph can be decomposed into a union of $k$ distinct isomorphic subgraphs. Recently, the work of [EEMM24] studied the smooth $k$-anonymity problem on a binary dataset, where they provide a detailed discussion comparing $k$-anonymity and differential privacy. In particular, in [EEMM24] the authors formally prove the following:

**Theorem 1.4.** *Let $\mathcal{M}$ be an arbitrary mechanism that satisfies $\epsilon$-edge differential privacy. Then, in order to achieve $E[J(\mathcal{M}(G), G)] \geq \alpha$, it must hold that $\epsilon = \Omega(\log(\alpha^2 nm))$, where $J(\cdot, \cdot)$ denotes the Jaccard similarity.*

This result suggests that achieving high utility (i.e., preserving the structure of the original graph) under differential privacy requires a large value of $\epsilon$. However, when $\epsilon$ is large, the algorithm likely maintains the graph $G$ unmodified thus exposing users to re-identification risks. Recall that the guarantee provided by $\epsilon$-DP is: $\Pr[\mathcal{M}(G) \in A] \leq e^\epsilon \Pr[\mathcal{M}(G') \in A]$, which becomes nearly vacuous for large $\epsilon$.

In contrast, $k$-anonymity offers a different privacy-utility tradeoff. Prior work shows that if the optimal anonymized graph $E_{opt}$ satisfies $J(E, E_{opt}) \geq 1 - O(1/\log k)$, then efficient algorithms can find solutions with comparable utility. Furthermore, in the case of smooth $k$-anonymity, where edge additions and deletions are allowed, the required assumption can be relaxed to $J(E, E_{opt}) \geq O(1)$.

## 2    Preliminaries

**Notation.**    In the $k$-anonymity problem, we are given a table $T$ having $n$ records, each with $d$ attributes. A record $r_i \in T$ is drawn from $\Sigma^d$, where $\Sigma$ is a finite set of possible values for each attribute. Then $r_i[j]$ is the value of the $j$-th attribute in $r_i$. Let $\star$ be a symbol not in $\Sigma$. Given a record $r \in \Sigma^d$, let its binary expansion be $x \in \{0, 1\}^{d|\Sigma|}$, where for $i \in [d], j \in [|\Sigma|]$, $p_{i,j} = 1$ if and only if the $i$-th attribute corresponds to the $j$-th value in $\Sigma$. Given two records $x, y \in \Sigma^m$, let their $\ell_0$ distance be $\text{dist}_{\ell_0}(x, y)$, which is the number of attributes for which $x$ and $y$ differ.

Given two vectors $x, y \in \mathbb{R}^m$, their $\ell_2$ distance is $\text{dist}_{\ell_2}(x, y) = \sqrt{\sum_{i=1}^m (x_i - y_i)^2}$. Given a finite point set $P \subseteq \mathbb{R}^m$, let $\rho_k(r)$ be the distance between $r$ and its $k$-th nearest neighbor in $P$ in $\ell_2$ distance. Specifically, if $r \in P$ we let $\rho_1(r) = 0$ (i.e., $r$'s 1-st nearest neighbor is $r$ itself).

## 2.1  $k$-Anonymity

**Definition 2.1** ($k$-Suppression Function). A $k$-suppression function $f$ maps each $r_i \in T$ to $r'_i$, by replacing some attributes of $r_i$ by $\star$. Moreover, for every $r \in T$, there exist $k-1$ other $r_1, r_2, \ldots, r_{k-1} \in T$ such that $f(r) = f(r_1) = f(r_2) = \cdots = f(r_{k-1})$. Define $c(f)$ to be the cost of $f$ on $T$, i.e., the number of attributes in $T$ replaced by $f$, where note that if the same attribute is changed in multiple records, its contribution to the cost is the number of records it is changed in.

**Definition 2.2** ($k$-anonymity via Suppression). In the $k$-anonymity problem, we are given a table of $n$ records and an anonymity parameter $k$. Our goal is to obtain a $k$-suppression function $f$ so that $c(f)$ is minimized. Specifically, we say $f$ is a $C$-approximation if $c(f) \leq C \cdot \min_{f'} c(f')$ .

## 3  Nearly-Linear Time Algorithm for $k$-Anonymity

In this section, we present our nearly-linear time approximation algorithm for $k$-anonymity. At a high level, we will first show that achieving an $O(k)$ approximation for $k$-anonymity can be reduced to solving the minimum-size constrained clustering problem with an $O(1)$ pointwise guarantee under the squared $\ell_2$ distance metric. Then, we will give an algorithm that solves this problem in nearly-linear time with high probability.

### 3.1  Reduction to Minimum Size Constrained Clustering

Recall that we have $n$ records $r_i$ ($i = 1, 2, \ldots, n$) in table $T$. For each record $r_i \in \Sigma^d$, let $x_i \in \{0, 1\}^{d|\Sigma|}$ be its binary expansion (i.e., a one-hot encoding of $r_i$, see definition in Section 2) and let $S$ denote the set of binary expansions of all records. Then for each pair of records $(r_i, r_j)$, the number of differing attributes between $r_i$ and $r_j$ is given by:

$$\text{dist}_{\ell_0}(r_i.r_j) = \frac{1}{2}\text{dist}_{\ell_0}(x_i, x_j) = \frac{1}{2}\text{dist}_{\ell_2}(x_i, x_j)^2 \ .$$

For each record $r_i$, let $r_j$ be its $k$-th nearest neighbor in $T$ with respect to the $\ell_0$ distance (i.e., the number of attributes on which the records differ, and recall that in our definition, the 1-st nearest neighbor of $r_i$ is $r_i$ itself). Consider an arbitrary partition of the $k$-anonymity problem on $T$. Since the group containing $r_i$ has at least $k$ records, the number of attributes that need to be suppressed for $k$-anonymity is at least $\text{dist}_{\ell_0}(r_i, r_j) = \frac{1}{2}\rho_k(x_i)^2$ . In the following lemma, we demonstrate that if there exists a partition $\mathcal{P} = \{P_1, P_2, \ldots, P_t\}$ with centers $c(P_1), c(P_2), \ldots, c(P_t)$ on $S$, such that for every point $p \in S$ in group $P_j$, the squared $\ell_2$ distance $\text{dist}_{\ell_2}(p, c(P_j))^2$ is at most $O(1) \cdot \rho_k(p)^2$, then this partition gives an $O(k)$-approximation to the $k$-anonymity problem on $T$. Formally, we have

**Lemma 3.1.** *Suppose that* $\mathcal{P} = \{P_1, P_2, \ldots, P_t\}$ *is a partition on $S$ with centers* $c(P_1), c(P_2), \ldots, c(P_t)$ *such that for every $P_i$, $k \leq |P_i| \leq 2k - 1$ and for every $p \in P_i$,*

$$\text{dist}_{\ell_2}(p, c(P_i))^2 \leq C \cdot \rho_k(p)^2 \ ,$$

*for some constant $C$. Then, the partition $\mathcal{P}$ is an $O(k)$-approximate solution for the $k$-anonymity problem on $T$.*

*Proof.* Let $\mathcal{Q} = \{Q_1.Q_2, \cdots, Q_s\}$ with centers $c(Q_1), c(Q_2), \cdots, c(Q_s)$ be the optimal solution to the $k$-anonymity problem on $T$. For each point $p \in P_i$, let $N(p, P_i)$ denote the number of attributes we need to suppress for $p$ with group $P_i$.

We first upper bound the total number of suppressed attributes over the partition $\mathcal{P}$. Given a point $p \in P_i$, recall that $p$ is a one-hot encoding of some record in table $T$ and let $r(p) \in T$ be the record that $p$ corresponds to. We have

$$
\begin{aligned}
N(p, P_i) &\leq \sum_{q \in P_i, q \neq p} \text{dist}_{\ell_0}(r(p), r(q)) \leq \sum_{q \in P_i, q \neq p} \frac{1}{2} \cdot \text{dist}_{\ell_0}(p, q) = \sum_{q \in P_i, q \neq p} \frac{1}{2} \cdot \text{dist}_{\ell_2}(p, q)^2 \\
&\leq \sum_{q \in P_i, q \neq p} \left( \text{dist}_{\ell_2}(p, c(P_i))^2 + \text{dist}_{\ell_2}(q, c(P_i))^2 \right)
\end{aligned}
$$

The first inequality is due to the fact that if we need to hide the attribute $j$, then there must be at least one $r(q)$ in $T$ such that the $j$-th attribute of $r(p)$ and $r(q)$ differ. Taking a sum over all $P_i$ and $p \in P_i$,

and noting that we have $|P_i| \leq 2k - 1$, we get that

$$\sum_{P_i \in \mathcal{P}} \sum_{p \in P_i} N(p, P_i) \leq (4k - 4) \sum_{P_i \in \mathcal{P}} \sum_{p \in P_i} \text{dist}_{\ell_2}(p, c(P_i))^2 .$$

This implies

$$\sum_{P_i \in \mathcal{P}} \sum_{p \in P_i} N(p, P_i) \leq (4k - 4) \cdot C \sum_{p \in S} \rho_k(p)^2 \tag{1}$$

from the assumption of the clustering solution. We next turn to lower bound the total number of hidden attributes over the partition $\mathcal{Q}$. Give a $q \in Q_i$, let $p$ denote its $k$-th nearest neighbor in $S$. Then we have

$$N(q, Q_i) \geq \max_{p' \in Q_i} \frac{1}{2} \cdot \text{dist}_{\ell_0}(p', q) .$$

Since there are at least $k$ points in $Q_i$, we have

$$N(q, Q_i) \geq \max_{p' \in Q_i} \frac{1}{2} \text{dist}_{\ell_0}(p', q) \geq \frac{1}{2} \text{dist}_{\ell_0}(p, q) = \frac{1}{2} \rho_k(q)^2$$

The last equation holds because both $p$ and $q$ are one-hot encodings of some records. Taking a sum over all $Q_i$ and $q \in Q_i$, we get that

$$\frac{1}{2} \sum_{q \in S} \rho_k(q)^2 \leq \sum_{Q_i \in \mathcal{Q}} \sum_{q \in Q_i} N(q, Q_i) \tag{2}$$

Combining (1) and (2) we get that

$$\sum_{P_i \in \mathcal{P}} \sum_{p \in P_i} N(p, P_i) \leq (8k - 8) \cdot C \sum_{Q_i \in \mathcal{Q}} \sum_{q \in Q_i} N(q, Q_i) ,$$

which is what we need. $\qquad\square$

We next note that the condition $|P_i| \leq 2k - 1$ can effectively be removed. If a group $P_i$ contains more than $2k - 1$ points, it can be divided into multiple smaller groups arbitrarily, each satisfying the condition.

**Corollary 3.2.** *Suppose that $\mathcal{P} = \{P_1, P_2, \ldots, P_t\}$ is a partition on $S$ with centers $c(P_1), c(P_2), \ldots, c(P_t)$ such that for every $P_i$, $|P_i| \geq k$ and for every $p \in P_i$,*

$$\text{dist}_{\ell_2}(p, c(P_i))^2 \leq C \cdot \rho_k(p)^2 ,$$

*for some constant $C$. Then the partition $\mathcal{P}$ can be efficiently transferred to another partition $\mathcal{P}'$, which is an $O(k)$-approximate solution to the $k$-anonymity problem on $T$.*

Finally, since the input to this clustering problem is the binary expansion of each record $r_i$ (which is in $d|\Sigma|$ dimensions) but not the record itself, naïvely it yields a $|\Sigma|$ factor in time and space, which can be large in practice. However, note that in the entire proof of Corollary 3.2, all we care about are the pair-wise distances. Consequently, we can use the following Johnson-Lindenstrauss lemma to reduce the dimension of each point to $O(\log n)$. Recall that since each of the input binary expansions is $d$-sparse, we can compute each embedding $\Phi x_i$ in $O(d \log n)$ time.

**Lemma 3.3** (Johnson-Lindenstrauss lemma, [JLS86])**.** *Let $\Phi \in \mathbb{R}^{r \times d}$ be a matrix whose entries are i.i.d samples from $\mathcal{N}(0, 1/r)$. For every vector $u \in \mathbb{R}^d$ and $\varepsilon \in (0, 1)$, we have $\Pr[(1 - \varepsilon)\|u\|_2 \leq \|\Phi u\|_2 \leq (1 + \varepsilon)\|u\|_2] \geq 1 - \exp(\Omega(\varepsilon^2 r)) .$*

### 3.2 Solving the Minimum Size Constrained Clustering Problem

After establishing Corollary 3.2, our goal shifts to finding a partition $\mathcal{P} = \{P_1, P_2, \ldots, P_t\}$ on an $O(\log n)$-dimensional pointset $S$ with centers $c(P_1), \ldots, c(P_t)$ such that $|P_i| \geq k$, and for every $p \in P_i$,

$$\text{dist}_{\ell_2}(p, c(P_i))^2 \leq C \cdot \rho_k(p)^2 .$$

In the remainder of this section, we shall present an algorithm that solves this problem in time $\widetilde{O}(nd + n \cdot (n/k)^{1/C^2 + o(1)})$. Our algorithm is inspired by the work of [EMMZ22] that studies this problem in the MPC setting, and their algorithm is not hard to adapt into an $\widetilde{O}(nd + n^{1+1/C^2+o(1)} \cdot k)$-time algorithm. This is at least $nk$ time, which can be as large as $n^2$ for $k = \Theta(n)$. We will significantly improve upon this runtime by using random sampling to reduce $k$-th nearest neighbor computations to 1-st nearest neighbor computations, described below.

We need the definition of locality sensitive hashing (LSH):

**Lemma 3.4** ([AI08, And09]). *Let $S = \{p_1, p_2, \cdots, p_n\} \subset \mathbb{R}^d$. Given two parameters $R > 0$ and $C > 1$, there is a hash family $\mathcal{H}$ such that $\forall p, q \in S$:*

1. *If $\|p - q\|_2 \leq R$, then $\Pr_{h \in \mathcal{H}}[h(p) = h(q)] \geq \mathcal{P}_1$ where $\mathcal{P}_1 \geq 1/n^{1/C^2 + o(1)}$.*

2. *If $\|p - q\|_2 \geq c_u \cdot C \cdot R$, then $\Pr_{h \in \mathcal{H}}[h(p) = h(q)] \leq \mathcal{P}_2$ where $\mathcal{P}_2 \leq 1/n^4$ and $c_u > 1$ is a universal constant.*

*Moreover, each hash function can be generated and evaluated in $n^{o(1)} d$ time.*

We now present our algorithm. At a high level, our algorithm can be divided into the following steps:

1. Sample a random subset $J \subseteq S$ with $|J| = O((n \log n)/k)$. For each $p_i \in J$, compute an $O(1)$-approximation to $\rho_k(p_i)$. Denote this distance by $d_i$. Also compute a set $N_i$, where $|N_i| \geq k - 1$ and for every $q \in N_i$ we have $\text{dist}_{\ell_2}(p_i, q) \leq O(1) \cdot \rho_k(p_i)$ (Lemma 3.6).

2. For each point $q$ not in $J$, find a $f(q)$ in $J$ satisfying $\text{dist}_{\ell_2}(q, f(q)) \leq O(1) \cdot \min_{p \in J} \text{dist}_{\ell_2}(q, p)$. For each $p \in J$, define $F(p) = \{q \in S \setminus J \mid f(q) = p\}$ (Lemma 3.7).

3. For $R_j = 3^j$, for $j = 1, 2, \ldots, O(\log d)$:

   (a) Let $A \subseteq J$ be the set of point $p_i$ where $R_{j-1} < \widetilde{\rho}_k(p_i) \leq R_j$ and $p_i$ has not been assigned.

   (b) Let $B \subseteq A$ be the set of point $p_i$ where none of the points in $N_i$ has been assigned in the previous iteration.

   (c) Greedily find one maximal independent set $C$ of $B$ such that the points in $C$ do not share the same point in their neighborhood set $N_i$. For each point $p_i \in C$, create a new cluster centered at $p_i$, assign points in $N_i$ to $p_i$.

   (d) For each point $p$ in $B \setminus C$, assume it shares the same neighbor with the point $s \in C$, assign $p$ to center $s$.

   (e) For each point $p_i$ in $A \setminus B$, assume one of its neighbor in $N_i$ has been assigned to center $s$, assign $p_i$ to $s$.

   (f) Furthermore, suppose that $p \in J$ has been assigned to center $s$, assign all unassigned points $q$ in $F(p)$ to the same cluster if $\text{dist}_{\ell_2}(q, p) \leq R_j$ (check in each iteration).

It is clear that each cluster we create during this procedure has size at least $k$. Moreover, each point $S$ during this procedure will be assigned to one cluster. To prove the correctness of the algorithm, we next present the following lemmas.

**Lemma 3.5.** *Given a set $J'$ with size $|J'| = O(n/k)$ and a distance parameter $R$, we can preprocess the set $J \cup J'$ in time $\log n \cdot (n/k)^{1 + 1/C^2 + o(1)}$ and then for every point $p \in J$, with probability $1 - 1/n^2$ we can compute a set $I$ such that $(1)$ $I$ has size at least the number of points in $J'$ that are within distance $R$ from $p$, and $(2)$ for every point $q \in I$, we have $\text{dist}_{\ell_2}(p, q) \leq O(C) \cdot R$.*

*Proof.* We draw $s = \Theta\left(\frac{\log n}{\mathcal{P}_1}\right)$ independent LSH hash functions in Lemma 3.4 with $S = J \cup J'$ and parameters $R$ and $C$. Then, for a fixed $i \in [s]$ and every $q \in J'$, we add $q$ into $I$ if $h_i(p) = h_i(q)$. We next prove the correctness of the algorithm. Consider $p, q \in J \cup J'$ with $\|p - q\|_2 \geq 2c_u \cdot C \cdot R$. Then for a fixed $i \in [s]$, $\Pr[h_i(p) = h_i(q)] \leq 1/(n/k)^4$ by Lemma 3.4. By taking a union bound over all such pairs of $\{p, q\}$ and all $i \in [s]$, with probability at least $1 - k/n$, we have for any $\{p, q\} \in S$ with $\|p - q\|_2 \geq 2c_u \cdot C \cdot R$, $h_i(p) \neq h_i(q)$ for all $i \in [s]$. Thus, if a point $q \in I$, we have $\|p - q\|_2 \leq 2c_u \cdot C \cdot R = O(C) \cdot R$. Now consider two points $p, q \in S$ with $\|p - q\|_2 \leq R$. By Lemma 3.4, with probability at least $1 - 1/(n/k)^3$, there exists an $i \in [s]$ such that $h_i(p) = h_i(q)$. By taking a union bound over all $\{p, q\}$ with $\|p - q\|_2 \leq R$, with probability at least $1 - k/n$, we have that $q \in I$ for all such pairs $\{p, q\}$. Finally, note that the above procedure only has a success probability of at least $1 - k/n$, but we can run the same procedure $O(\log n)$ independent times to boost the success probability to $1 - 1/n^2$ (after obtaining $I$, we can check whether $I$ satisfies the condition or not by computing the pairwise distances). $\square$

**Lemma 3.6.** *We can preprocess the point set $S$ and $J$ in time $k \cdot (n/k)^{1 + 1/C^2 + o(1)}$, and after that for every point $p \in J$, we can with probability at least $1 - 1/n^2$ compute an $O(C)$-approximation $\widetilde{\rho}_k(p)$ to $\rho_k(p)$ in time $k \cdot (n/k)^{1/C^2 + o(1)}$ with set $N_i$ such that $|N_i| \geq k - 1$ and for every $q \in N_i$, we have $\text{dist}_{\ell_2}(p, q) \leq \widetilde{\rho}_k(p)$.*

*Proof.* The procedure is defined as follows. We split $S/J$ into $m = O(k)$ disjoint subsets $S/J = J_1 \cup J_2 \cup \ldots J_m$ with each $|J_i| = n/k$. Let $R_i = 2^i$. For each $R_i$ ($i = 0, 1, \cdots, O(\log d)$) and every $j \in [m]$, we run the procedure in Lemma 3.5. For a point $p \in J$, let $N_i^j$ be the subset returned by Lemma 3.5 with distance parameter $R_i$, and set $J' = J_j$ and specifically, let $N_i^0$ be the subset returned by Lemma 3.5 with the set $J$ itself. Let $R_i$ be the smallest $i$ for which $|\bigcup_{j=0}^m N_i^j| \geq k - 1$. We use $R_i$ as an approximation to $\rho_k(p)$ and return the set $N_i = \bigcup_{j=0}^m N_i^j$.

We next prove the correctness of our algorithm. Let $i$ be the integer for which $R_{i-1} < \rho_k(p) \leq R_i$. This means there are at least $k - 1$ points within distance $R_i$ from $p$. From the guarantee of Lemma 3.5 we have that with probability $1 - 1/n^2$, $|\bigcup_{j=0}^m N_i^j| \geq k - 1$. On the other hand, for an $R_{i'} \leq \rho_k(p)/O(C)$, from the guarantee of Lemma 3.5 we have with probability $1 - 1/n^2$, we have $|\bigcup_{j=0}^m N_{i'}^j| < k - 1$. Moreover, similar to Lemma 3.5, we have that after taking a union bound, with probability at least $1 - 1/n^2$, for every point $q \in N_i$, $\mathrm{dist}_{\ell_2}(p, q) \leq O(C) \cdot \rho_k(p)$.

Finally, we consider the time complexity of the algorithm. Note that we do not need to explicitly compare the hash value of each pair $\{p, q\}$. Instead for the point $p$ we care about, we can just look at the cell it falls in for each of the hash functions. Moreover, we can terminate the procedure and return $N_i$ after the set $N_i$ we maintain has size $k - 1$. Hence, the overall runtime for one point $p \in J$ is $k \cdot (n/k)^{1/C^2 + o(1)}$. □

**Lemma 3.7.** *Let $Q$ be a subset of $J$ with size $O(n \log n/k)$. We can pre-process $Q$ and the point set $S$ in time $k \cdot (n/k)^{1+1/C^2+o(1)}$, such that afterwards, given a point $p \in S$, with probability at least $1 - 1/n^2$ we can find a point $s_j \in Q$ in time $(n/k)^{1/C^2+o(1)}$ such that $\mathrm{dist}_{\ell_2}(p, s_j) \leq O(1) \cdot \min_{s_i \in Q} \mathrm{dist}_{\ell_2}(p, s_i)$.*

*Proof.* Note that we have $|Q| \leq O(n \log n/k)$. Similarly to what we do in Lemma 3.6, we split $S$ into $m = O(k)$ disjoint subsets $S = J_1 \cup J_2 \cup \ldots J_m$ with each $|J_i| = O(n/k)$. Let $R_i = 2^i$. For every $R_i$ ($i = 0, 1, \cdots, O(\log d)$) and every $j \in [m]$, we run the procedure in Lemma 3.5 on $Q \cup J_j$. For a point $p \in S$. Let $R_i$ be the smallest integer such that there exists an $N_i^j$ such that $N_i^j \cap Q \neq \emptyset$, and the algorithm will return one arbitrary center $s_\ell$ in $N_i^j \cap Q$. Similar to the proof of Lemma 3.6, we have that this $s_\ell$ satisfies $\mathrm{dist}_{\ell_2}(p, s_\ell) \leq O(1) \cdot \min_i \mathrm{dist}_{\ell_2}(p, s_i)$, which is what we need. □

**Lemma 3.8.** *For every point $p_i \in J$, if $q \in N_i$, then we have $\rho_k(q) \leq O(C) \cdot \rho_k(p)$. Moreover, for every point $p \notin J$, with probability at least $1 - 1/n^2$, we have that $\rho_k(f(p)) \leq O(C) \cdot \rho_k(p)$.*

*Proof.* Let $N_i' = \{q_1', q_2', \ldots, q_{k-1}'\}$ denote the set of $p_i$'s true $k$-nearest neighbors. Note that from the property of $N_i$, we have that $\mathrm{dist}_{\ell_2}(p, q_j) \leq O(C) \cdot \rho_k(p)$. Consider an arbitrary $q_j \in N_i$ we have that for other $q_\ell' \in N_i'$, $\mathrm{dist}_{\ell_2}(q_j, q_\ell') \leq \mathrm{dist}_{\ell_2}(p, q_j) + \mathrm{dist}_{\ell_2}(p, q_{\ell'}) \leq O(C + 1) \cdot \rho_k(p)$. This implies $\rho_k(q_j) \leq O(C + 1) \cdot \rho_k(p)$. Moreover, for each $p \notin J$, since $J$ has size $O(n \log n/k)$, we have that with probability at least $1 - 1/n^2$ there exists a $q_j' \neq p$ such that $q_j' \in J$. This implies that $\min_{q \in J} \mathrm{dist}_{\ell_2}(p, q) \leq \rho_k(p)$. Then we have $\mathrm{dist}_{\ell_2}(p, f(p)) \leq O(C) \cdot \min_{q \in J} \mathrm{dist}_{\ell_2}(p, q) \leq O(C) \cdot \rho_k(p)$. Then, from a similar argument we can get $\rho_k(f(p)) \leq O(C + 1) \cdot \rho_k(p)$. □

**Lemma 3.9.** *Suppose that in phase $j$ the point $p$ is assigned to $s$, then we have $\mathrm{dist}_{\ell_2}(p, s) \leq 3 \cdot R_j$.*

*Proof.* The proof is by induction. For $j = -1$, i.e., before the phase of $j = 0$, since there is no point assigned at this phase, the lemma statement automatically holds. Now consider the $i$-th phase. We suppose that every assignment before the $i$-th phase has radius at most $3 \cdot R_{j-1} = R_j$.

We first consider the assignments in step 3(c). In this case, for each point $p_i$ in $C$, since $R_j \leq \widetilde{\rho}_k(p_i) \leq R_j$, the distance from $p_i$ to the points in its neighbor set $N_i$ will be at most $R_j$, which means this assignment has distance at most $R_j$. We next consider the assignment in step 3(d). In this case, since $C$ is a maximal independent set, we have that for each point $p$ in $B \setminus C$, it shares the same neighbor $q$ with points $s \in C$. This implies $\mathrm{dist}_{\ell_2}(p, s) \leq \mathrm{dist}_{\ell_2}(p, q) + \mathrm{dist}_{\ell_2}(q, s) \leq R_j + R_j = 2 \cdot R_j$.

We next consider the assignment in step 3(e). For each point $p_i$ in $A \setminus B$, we have that there exists one $q \in N_i$ has been assigned to center $s$ in the previous iterations with distance at most $3 \cdot R_{j-1} = R_j$. We have $\mathrm{dist}_{\ell_2}(p_i, s) \leq \mathrm{dist}_{\ell_2}(p_i, q) + \mathrm{dist}_{\ell_2}(q, s) \leq R_j + R_j = 2 \cdot R_j$.

We finally consider the assignment in step 3(f). Suppose we assign $p \notin J$ to center $s$, then we have we also assign $f(p)$ to $s$. Then we have that $\mathrm{dist}_{\ell_2}(p, s) \leq \mathrm{dist}_{\ell_2}(p, f(p)) + \mathrm{dist}_{\ell_2}(f(p), s) \leq R_j + 2 \cdot R_j \leq 3 \cdot R_j$. □

---
**Algorithm 1** $k$-Anonymity via Near Neighbors
---
1: **Input:** A table $T$ that contain $n$ records, parameters $k \geq 1$.
2: Let $\Phi \in \mathbb{R}^{r \times d|\Sigma|}$ be the JL matrix in Lemma 3.3. For each record $r_i \in T$, compute $p'_i = \Phi \cdot p_i$ where $p_i$ is a binary expansion of $r_i$. Let $S = \{p'_1, p'_2, \ldots, p'_n\}$.
3: Use Lemma 3.11 obtain a partition $\mathcal{P} = \{P_1, P_2, \ldots, P_t\}$ on $S$.
4: For each $i \in [t]$, if $|P_i| \leq 2k - 1$, let $\mathcal{P}_i = \{P_i\}$. Otherwise split $P_i = Q_{i,1} \cup Q_{i,2}, \cdots \cup Q_{i,\ell}$ where $k \leq |Q_i| \leq 2k - 1$ and let $\mathcal{P}_i = \{Q_{i,1}, Q_{i,2}, \ldots, Q_{i,\ell}\}$.
5: Return the partition $\mathcal{Q} = \bigcup_{i=1}^{t} \mathcal{Q}_i$.
---

**Lemma 3.10.** *For every $p \in S$, with probability at least $1 - 1/n^2$, $p$ is assigned with a distance to its center at most $O(C^3) \cdot \rho_k(p)$.*

*Proof.* We first consider the point $p \in J$. Let $j$ be the number such that $R_{j-1} < \widetilde{\rho}_k(p) \leq R_j$. Then, from the procedure of the algorithm, we have that $p$ must be assigned in or before the phase $j$. From Lemma 3.9 we have the distance from $p$ to its center will be at most $3 \cdot R_j \leq 9 \cdot \widetilde{\rho}_k(p) \leq O(9C) \cdot \rho_k(p)$.

We next consider the points $p \notin J$. Let $j$ be the number such that $R_{j-1} < O(C^2) \cdot \widetilde{\rho}_k(p) \leq R_j$. From Lemma 3.8 we have with probability at least $1 - 1/n^2$, we have $\rho_k(f(p)) \leq O(C) \cdot \rho_k(p)$ and $\text{dist}_{\ell_2}(p, f(p)) \leq O(C) \cdot \rho_k(p)$. Hence, from the procedure of our algorithm, we have that $p$ must be assigned in or before phase $j$, which implies the distance from $p$ to its center will be at most $3 \cdot R_j \leq O(9C^3) \cdot \rho_k(p)$. $\qquad\square$

By Lemma 3.6, Lemma 3.7, and Lemma 3.10, we get the correctness of the following lemma.

**Lemma 3.11.** *There is an algorithm, which outputs a partition $\mathcal{P} = \{P_1, P_2, \ldots, P_t\}$ on $S$ such that with high probability, for every $P_i$, $|P_i| \geq k$ and for every $p \in P_i$,*

$$\text{dist}_{\ell_2}(p, c(P_i))^2 \leq O(C^6) \cdot \rho_k(p)^2 \, ,$$

*for some constant $C$. Moreover, the entire procedure runs in time $\widetilde{O}\left(n \cdot (n/k)^{1/C^2 + o(1)}\right)$.*

*Proof of Theorem 1.1.* The entire algorithm is presented in Algorithm 1. We first prove the correctness of our algorithm. From Lemma 3.11 we have with high probability, the partition $\mathcal{P} = \{P_1, P_2, \ldots, P_t\}$ satisfies for every $P_i$, $|P_i| \geq k$ and for every $p \in P_i$. $\text{dist}_{\ell_2}(p, c(P_i))^2 \leq O(C^6) \cdot \rho_k(p)^2$ , Then, after splitting the subsets $|P_i| \geq 2k$, from Lemma 3.1 we have the partition $\mathcal{Q}$ is an $O(k)$-approximation solution to the $k$-anonymity on the table $T$. The overall time complexity is $\widetilde{O}\left(n \cdot (n/k)^{1/C^2 + o(1)}\right)$. $\qquad\square$

## 4 Single-Point $k$-Anonymity

We study the following single-point $k$-anonymity problem.

**Definition 4.1** (single-point $k$-anonymity)**.** In the single-point $k$-anonymity problem, we are given a table $T$ that contains $n$ records and a specific record $p \in T$. Then, we ask to choose $k - 1$ other records $r_1, r_2, \ldots, r_{k-1}$ from $T$ with the goal being to minimize the number of attributes the group $(p, r_1, r_2, \ldots, r_{k-1})$ has to be suppressed, i.e., the number of $j$ such that

$$\exists a, b \in \{p, r_1, r_2, \ldots, r_{k-1}\}, a[j] \neq b[j].$$

We say a solution is a $C$-approximation if the number of hidden (suppressed) attributes in this solution is at most $C$ times the number of hidden attributes in the optimal solution.

**Upper Bound.** We observe a straightforward method to achieve a $(k - 1)$-approximation: select the $i$-th nearest neighbor of $p$ in $T$ (with respect to $\ell_0$ distance, and excluding $p$ itself) for $i = 1, 2, \ldots, k - 1$. To see why this works, let $r_i$ denote $p$'s $i$-th nearest neighbor in $T$. On the one hand, we have that the number of hidden attributes in the optimal solution is at least $\text{dist}_{\ell_0}(p, r_{k-1})$. On the other hand, we have that the number of hidden attributes in the solution $(r_1, r_2, \ldots, r_{k-1})$ is at most $\sum_{i=1}^{k-1} \text{dist}_{\ell_0}(p, r_i) \leq (k - 1) \cdot \text{dist}_{\ell_0}(p, r_{k-1})$.

**Lemma 4.2.** *There is a deterministic algorithm that computes a $(k - 1)$-approximation of the single-point $k$-anonymity problem in time $O(nd + n \log n)$.*

**Lower Bound.** We next consider lower bounds for the single-point $k$-anonymity problem. In [CDK12], the authors give the following conjecture about the time complexity of the following DENSE VS RANDOM problem: given a graph $G$, it is hard to distinguish between the following two cases: (1) $G = G(n,p)$ where $p = n^{\alpha-1}$ (and thus the graph has log-density concentrated around $\alpha$), and (2) $G$ is adversarially chosen so that the densest $\ell$-subgraph has log density $\beta$ where $\ell^\beta \gg p\ell$ (and thus the average degree inside this subgraph is approximately $\ell^\beta$).

In [CDM17], the work studies the M$k$U problem and extends the conjecture to the hypergraph case: Given an $r$-uniform hypergraph $G$ on $n$ nodes, distinguish between the following two cases:(1) $G = G(n,p,r)$ where $p = n^{\alpha-(r-1)}$ (and thus the graph has log-density concentrated around $\alpha$), and (2) $G$ is adversarially chosen so that the densest $\ell$-subhypergraph on $\ell$ vertices and has log density $\beta$ where $\ell^\beta \gg p\ell$ (and thus the average degree inside this subhypergraph is approximately $\ell^\beta$).

**Conjecture 4.3.** *For all constant $r$ and $0 < \beta < r - 1$, for all sufficiently small $\varepsilon > 0$, and for all $\ell^{1+\beta} \leq n^{(1+\alpha)/2}$, we cannot solve* HYPERGRAPH DENSE VS RANDOM *with log-density $\alpha$ and planted log-density $\beta$ in polynomial time (w.h.p.) when $\beta \leq \alpha - \varepsilon$.*

Assuming the above conjecture, we prove the following hardness result. Our construction is based on the lower bound for the minimum $k$-union problem studied in [CDM17].

*Proof of Theorem 1.3.* We shall show that, if we have an algorithm for single-point $k$-anonymity with approximation ratio $k^{1-O(1/C)}$, then it can be used to solve the HYPERGRAPH DENSE VS RANDOM with the specific parameters in Conjecture 4.3.

For sufficiently large constant $r$, let $\alpha = \sqrt{r} - 1$ and $\beta = \sqrt{r} - 1 - \varepsilon$, $n = d$, and $\ell = d^{1/\sqrt{r}}$. Given an instance of the input hypergraph in Conjecture 4.3, we construct the input instance to the single-point $k$-anonymity problem as follows. First, we set the specific record $p$ to be a $d$-dimensional zero vector. Next, for the $i$-th edge in the hypergraph, we set the record $r_i$ to be the binary vector where its $j$-th coordinate is 1 if and only if the $j$-vertex is included in this edge. Let the table $T$ be the set that contains all $r_i$'s and $k = \Theta(d^{1-\varepsilon/\sqrt{r}})$.

Then, consider the $\ell$ hypersubgraph in case two. With high probability it will have $\Theta(\ell^{1+\beta}) = \Theta(\ell^{\sqrt{r}-\varepsilon}) = \Theta(d^{1-\varepsilon/\sqrt{r}})$ edges. Hence, setting $k = \Theta(d^{1-\varepsilon/\sqrt{r}})$ and choosing the records that correspond to these edges in the subhypergraph, the number of attributes we need to hide is at most $\ell = d^{1/\sqrt{r}}$ nodes in $G$. We next consider $G$ in case one. We claim that with high probability every $k$ edges in $G$ will cover at least $d^{1-1/\sqrt{r}+1/2r-\varepsilon/r^{3/2}}$ nodes in $G$. This means that every $k$ records in $T$ will need to have such a number of attributes hidden. To prove this, we only need to show that when $\widetilde{\ell} = d^{1-1/\sqrt{r}+1/2r-\varepsilon/r^{3/2}}$, with high probability for every $\widetilde{\ell}$ subhypergraph in $G$, this subhypergraph can only have at most $k - 1$ edges. To get this, note that the expectation of the number of edges in each of the subhypergraphs is on the order of $\widetilde{\ell}^r n^{\sqrt{r}-r} = d^{1/2-\varepsilon/\sqrt{r}}$. Then by Chernoff's bound we have with failure probability at most $2\exp(-d)$ that this subhypergraph has fewer than $d^{1-\varepsilon/\sqrt{r}}$ edge. Taking a union bound on the $\binom{d}{\widetilde{\ell}}$ subhypergraphs, we get the desired result.

The ratio of the two cases will be at least $\widetilde{\ell}/\ell = d^{1-2/\sqrt{r}+1/(2r)-\varepsilon/r^{3/2}}$ , which rules out algorithms for single-point $k$ anonymity with ratio $k^{1-O(1/\sqrt{r})}$. The constant $r$ here can be sufficiently large and in our construction the total number of edges will be $\Theta(n^r \cdot n^{\sqrt{r}-r}) = \Theta(d^{\sqrt{r}}) \leq k^{\sqrt{r}}$. $\square$

# 5 Experiments

All of our experiments were conducted on a device with a 3.30GHz CPU and 16GB RAM. We will use the following dataset which has been widely used in the study of anonymized privacy protection:

- **Adult.**[3] The Adult data contains 48842 tuples from US Census data. The tuples with missing values are removed. In particular, we choose 8 attributes as quasi-identifier.

We observe that in the dataset we use, most points have several neighbors with a small distance. Hence, in our implementation we use MinHash as an instance of LSH for simplification.

As a baseline, we consider the clustering-based heuristic algorithms proposed in [ZWL⁺18], where the authors demonstrate that their approach outperforms other existing heuristic methods, such as Mondrian [LDR06]. Although this algorithm shows strong performance on real-world datasets, it also

---

[3]The Adult from the UCI Machine Learning Repository.

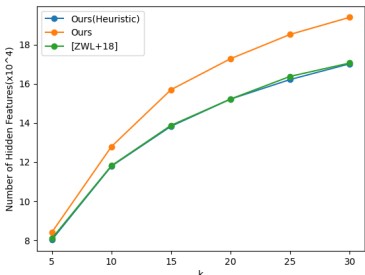

Figure 1: Test result for $k$-anonymity.

exhibits relatively high computational complexity. The core idea of the algorithm can be summarized as follows:

1. Iteratively selecting a new cluster center as the point with the highest average distance from existing centers, and

2. Assigning the $k-1$ nearest neighbors of that point to the same cluster. This strategy, while intuitive, results in at least $\Omega(n^2)$ time complexity, even when we disregard other parameters such as $d$ and $k$.

Motivated by this bottleneck, we investigated whether the core heuristic could be accelerated using ideas from our own algorithm. We found that substantial speed-ups are indeed possible, with minimal impact on performance. In particular, we introduce the following modifications:

1. For the center selection step (Step 1), instead of computing the average distance for all points, we randomly sample a fixed number of candidate points.

2. Once a new center is chosen, we leverage Locality-Sensitive Hashing (LSH) to efficiently compute its approximate $k$-nearest neighbors. This process is similar to the procedures in Lemmas 3.5 and 3.6. [4]

In our experiments, these modifications lead to a substantial reduction in runtime while preserving performance comparable to the original heuristic. We refer to the improved version of our new algorithm as Ours (Heuristic).

To empirically assess runtime performance, we conducted experiments comparing the two algorithms:

- **Ours (Heuristic)** : Implemented in C++ for efficiency.
- [ZWL+18]: Since no official implementation was available, we implemented the algorithm ourselves in C++. We applied the same optimization strategies as in our own implementation.

**Results Summary.** The experimental results are presented in Figure 1. We vary the value of $k$ from 5 to 30 and report the number of hidden attributes. As shown in the figure, our original approach performs similarly to [ZWL+18] when $k$ is small but exhibits worse performance as $k$ increases. In contrast, our second approach, which incorporates the heuristic, closely matches the performance of [ZWL+18] across the entire range of $k$.

We next evaluate the runtime of different algorithms on the Adult dataset for $k$ values ranging from 5 to 30. As shown in Table 2, our algorithmic ideas can be adapted to existing heuristic methods, leading to significant speed-ups while preserving comparable performance.

Table 2: Runtime comparison on Adult dataset (s)

| Dataset | ZWL+18 | Ours (Heuristic) |
|---|---|---|
| Adult ($k=10$) | 894.216 | 17.431 |
| Adult ($k=15$) | 394.386 | 5.914 |
| Adult ($k=20$) | 222.038 | 3.368 |

---

[4]In our experiments, we found that because the dataset is relatively small, this step does not significantly improve the runtime for this heuristic. Therefore, our current results only incorporate the first modification.

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

## A  Preliminaries

Given a graph $G(V, E)$, for any $v \in V$, let $\Gamma_G(v)$ be the set of the neighbors of $v$ in $G$. A maximal independent set of $G$ is a subset $I \subseteq V$ such that every vertex in the graph is at distance at most 1 from some vertex in $I$. A $\beta$-ruling set of a graph is an independent set $I$ such that every vertex in the graph is at a distance of at most $\beta$ from some vertex in $I$. In particular, a maximal independent is a 1-ruling set, which can be seen as a special case. Let $G^2$ to be the square graph of $G$ that has the same set of vertices as $G$, but in which two vertices are connected when their distance in $G$ is at most 2.

### A.1  Massivaly Parallel Computing

The MPC model [FMS$^+$10, KSV10, GSZ11, BKS17, ANOY14] is an abstract of modern massively parallel computing systems such as Map Reduce [DG08], Hadoop [Whi12], Dryad [IBY$^+$07], Spark [ZCF$^+$10] and others.

We follow the introduction in [EMMZ22]. In the MPC model, the input data has size $N$. The system consists of $p$ machines, each with a local memory size of $s$. Hence, the total space available in the entire system is $p \cdot s$. The space here is measured by words, each of $O(\log(p \cdot s))$ bits. Specifically, if the total space $p \cdot s = O(N^{1+\gamma})$ for some $\gamma \geq 0$ and the local space $s = O(N^\delta)$ for some $\delta \in (0, 1)$, then the model is referred to as the $(\gamma, \delta)$-MPC model [ASS$^+$18]. The computation proceeds in synchronized rounds. In each round, every machine processes the data stored in its local memory and sends messages to other machines at the end of the round. Although each machine can send messages to any other machine, the total size of the messages sent or received by a machine in a single round should be at most $s$.

Note that the space of each machine is sublinear in the input size. This means that we cannot collect all input data into one machine. An MPC algorithm is called *fully-scalable*, if it can work when the space per-machine is $O(N^\delta)$ for any constant $\delta \in (0, 1)$. The goal in this work is to design fully-scalable MPC algorithms minimizing the number of rounds while using a small total space.

## B  Extending to MPC Model

In this section, we demonstrate that our algorithm can be adapted into a fully scalable MPC algorithm. At a high level, [EMMZ22] studied the minimum size constraint clustering problem in the MPC model and give an algorithm that can be efficiently implemented in the MPC model. However, since the input of this clustering problem is the binary expansion of each record $r_i$ (which is in dimension $d|\Sigma|$) but not the record itself, naively using the algorithm in [EMMZ22] yields a $|\Sigma|$ factor in time and space, which can be large in practice. Instead, we open the procedure of this algorithm and show that since each input point is $d$-sparse, it is still achievable in the same order of time and space.

We will need the following concept of the $C$-approximate $(R, r)$-near neighbor graph.

**Definition B.1** ($C$-approximate $(R, k)$-near neighbor graph, [EMMZ22])**.** Consider a point set $P$ from a metric space $\mathcal{X}$. Let $C, R, k \geq 1$. If an undirected graph $G = (V, E)$ satisfies

(a) $V = P$,

(b) $\forall (p, p') \in E, \text{dist}_{\mathcal{X}}(p, p') \leq C \cdot R$,

(c) For every $p \in P$, either $|\Gamma_G(p)| \geq k$ or $\{p' \in P \mid \text{dist}_{\mathcal{X}}(p, p') \leq R\} \subseteq \Gamma_G(p)$,

then we say $G$ is a $C$-approximate $(R, k)$-near neighbor graph of $P$.

We will show that given parameters $C, R$ and a set $S$ where each $p \in S$ represents a binary expansion of one record $r_i$ in table $T$. We can efficiently build a $C$-approximate $(R, k)$-near neighbor graph $G$ of $S$ under $\ell_2$ distance in $\widetilde{O}(nd + n^{1+1/C^2+o(1)}k)$ time in the static setting and in $O(1)$ rounds in the MPC model.

**Lemma B.2.** *Given parameters $C, R$, and a set $S$ where each $p \in S$ is a binary expansion of a record $r_i$ in $T$. Then, there is an algorithm that with high probability constructs a $O(C)$-approximate $(R, k)$-near neighbor graph of $S$ in time $\widetilde{O}(nd + n^{1+1/C^2+o(1)}k)$. The size of $G$ is at most $\widetilde{O}\left(n^{1+1/C^2+o(1)} \cdot k\right)$.*

*Proof.* Let $\Phi \in \mathbb{R}^{r \times d|\Sigma|}$ be a JL matrix in Lemma 3.3 with $r = O(\log n)$ and for each $p \in S$, we compute $p' = \Phi p$. Let $S' = \{p'_1, p'_2, \ldots, p'_n\}$. The construction of the graph $G$ is as follows. We draw $s = \Theta\left(\frac{\log n}{\mathcal{P}_1}\right)$ independent LSH in Lemma 3.4 with parameters $R$ and $C$. Then, for every $i \in [s]$ and every $p' \in S'$, we connect $p'$ to $k$ arbitrary points $q' \in S'$ in $G$ with $h_i(p') = h_i(q')$. If there are less than $k$ points with $h_i(q') = h_i(p')$, connect $p'$ to all such $q'$ in $G$.

We first prove the correctness of the algorithm. From Lemma 3.3 we have with probability at least $1 - 1/n$, we have for every pair of $p'$ and $q'$ in $S'$, $0.9\|p - q\|_2 \leq \|p' - q'\|_2 \leq 1.1\|p-q\|_2$. Condition on this event occurs and consider $p, q \in S$ with $\|p-q\|_2 \geq 2c_u \cdot C \cdot R$. Then from the assumption we have $\|p' - q'\|_2 \geq 1.8c_u \cdot C \cdot R$, this means that for a fixed $i \in [s]$, $\Pr[h_i(p') = h_i(q')] \leq 1/n^4$ by Lemma 3.4. By taking union bound over all such pairs of $(p,q)$ and all $i \in [s]$, with probability at least $1 - 1/n$, we have for any $(p,q) \in S$ with $\|p - q\|_2 \geq 2c_u \cdot C \cdot R$, $h_i(p') \neq h_i(q')$ for all $i \in [s]$. Thus, if an edge $(p,q) \in E$, we have $\|p-q\|_2 \leq 2c_u \cdot C \cdot R = O(C) \cdot R$. Now consider two points $p, q \in S$ with $\|p-q\|_2 \leq R$. From the assumption we have $\|p' - q'\| \leq 1.1 \cdot R$, By Lemma 3.4 and Chernoff bound, with probability at least $1 - 1/n^3$, there exists an $i \in [s]$ such that $h_i(p') = h_i(q')$. By taking union bound over all $(p,q)$ with $\|p - q\|_2 \leq R$, with probability at least $1 - 1/n$, we have there is an edge $(p,q) \in E$ for all such pairs $(p,q)$. Combining these two aspects, we have that for every $p \in P$, either $\{q \in P \mid \|p - q\|_2 \leq R\} \subseteq \Gamma_G(p)$ or $|\Gamma_G(p)| \geq k$.

We next consider the runtime complexity. First, for each $p \in S$, since $p$ is $d$-sparse, we can compute $\Phi \cdot p$ is $O(d \log n)$ time, which implies we can form the set $S'$ in $O(nd \log n)$ time (note that the algorithm does not need to write down $p$ explicitly). Then from Lemma 3.4 we get that algorithm can evaluate $h_i(p')$ for every $i \in [s]$ and $p \in S'$ in time $O(n^{1+1/C^2+o(1)})$ as the dimension of each $p' \in S$ is $d' = O(\log n)$. Finally, to connect edges in graph $G$, we sort points in $S'$ via their hash values and only consider to connect the points with the same hash values. Since for each hash function, we connect at most $r$ edges from a point, we have this procedure can be done in time $\widetilde{O}\left(n^{1+1/C^2+o(1)} \cdot k\right)$ and the size of $G$ is at most $\widetilde{O}\left(n^{1+1/C^2+o(1)} \cdot k\right)$. $\square$

**Lemma B.3.** *Given parameters $R, C, k$ and point set $S$. There is an MPC algorithm that builds an $O(C)$-approximate $(R,r)$-near neighbor graph of $S$ with high probability in $O(1)$ rounds and $\widetilde{O}\left(nd + n^{1+1/C^2+o(1)}k\right)$ total space.*

*Proof.* We first note that, the Johnson-Lindenstrauss lemma can be implemented in $O(1)$ MPC round and $O(ndr)$ space where $r = O(\log n)$ in our case (See, e.g., Appendix A in [EMMZ22]). Next, we can handle LSH functions in parallel. According to Lemma 4.1, we use $O(1)$ rounds to compute LSH values for all points in $S$. To connect edges, we can sort points via their LSH values, make copies of some vertices and query indices in parallel. These operations can be done simultaneously in $O(1)$ rounds [Goo96, GSZ11, ASS$^+$18]. Since we run $s$ independent LSH functions and for each $i \in [s]$, every each connects to at most $r$ vertices, we have the total space needed is $\widetilde{O}(nd + n^{1+1/C^2+o(1)}k)$. $\square$

Given access to $O(C)$-approximate $(R, k)$-near neighbor graphs with different distance parameters $R_i = 2^i$ for $i = 0, 1, \cdots, O(\log d)$, the MPC algorithm presented in [EMMZ22] produces a partition $\mathcal{P}$ on $S'$ that satisfies the condition outlined in Corollary 3.2.

**Lemma B.4** (Essentially Theorem 4.31 in [EMMZ22]). *Let $\gamma, \varepsilon \in (0,1)$. Given the $C$-approximate $(R_i, k)$-near neighbor graph $G_i$'s, there is a fully scalable MPC algorithm that with high probability outputs a partition $\mathcal{P} = \{P_1, P_2, \ldots, P_t\}$ on $S$ such that with high probability, for every $P_i$, $|P_i| \geq k$ and for every $p \in P_i$, $\mathrm{dist}_{\ell_2}(p, c(P_i))^2 \leq O\left(\frac{\log^2(1/\varepsilon)}{\gamma}\right) \cdot \rho_k(p)^2$. The algorithm takes $O\left(\frac{\log 1/\varepsilon}{\gamma} \cdot \log^{1+\varepsilon}(n) \log\log(n)\right)$ parallel time and $\widetilde{O}\left(n^{1+\gamma+o(1)} \cdot k\right)$ total space.*

The full algorithm of Lemma B.4 is presented in Algorithm 2 for completeness.

---

**Algorithm 2** Clustering with Pointwise Guarantee

---

1: **Input:** A point set $P$, a parameter $k \geq 1$.
2: Let $C \geq 1$.
3: Let $t \leftarrow 0$. Initialize the family of clusters $\mathcal{P} \leftarrow \emptyset$.
4: Let $\Delta$ $(\delta)$ be an upper bound (a lower bound) of $\mathrm{dist}_{\ell_2}(p, q)$ for $p \neq q \in P$.
5: Let $L = \lceil \log(\Delta/\delta) \rceil$. For $i \in \{0, 1, 2, \cdots, L\}$, let $R_i \leftarrow 2^i \cdot \delta$.
6: **for** $i = 0 \to L$ **do**
7:     Compute a $C$-approximate $(R_i, r)$-near neighbor graph $G_i = (P, E_i)$ of $P$.
8:     Let $P_i' \subseteq P$ be the vertices with at least $k$ neighbors in $G_i$, i.e., $P_i' = \{p \in P \mid |\Gamma_{G_i}(p)| \geq k\}$.
9:     Let $P_i'' = \left\{ p \in P_i' \mid \mathrm{dist}_{G_i}\left(p, \bigcup_{Q \in \mathcal{P}} Q\right) > 1 \right\}$.
10:     Compute a $\beta$-ruling set $S_i = \{s_{i,1}, s_{i,2}, \cdots, s_{i,t_i'}\}$ of $(G_i^2)[P_i'']$.
11:     Compute $P_i''' = \left\{ p \in P \setminus \bigcup_{Q \in \mathcal{P}} Q \mid \mathrm{dist}_{G_i}(p, S_i) \leq 2 \cdot \beta \right\}$.
12:     Partition $P_i'''$ into $t_i'$ clusters $Q_{i,1}, Q_{i,2}, \cdots, Q_{i,t_i'}$ where the center $c(Q_{i,j})$ is $s_{i,j}$. For each point $p \in P_i''' \setminus S_i$, add $p$ into an arbitrary cluster $Q_{i,j}$ such that $\mathrm{dist}_{G_i}(p, s_{i,j})$ is minimized.
13:     For each $p \in P_i' \setminus P_i'''$, if $p \notin \bigcup_{Q \in \mathcal{P}} Q$, find an arbitrary cluster $Q \in \mathcal{P}$ such that $\mathrm{dist}_{G_i}(p, Q) \leq 1$ and update $Q$ by adding $p$ into $Q$.
14:     Add $Q_{i,1}, Q_{i,2}, \cdots, Q_{i,t_i'}$ into $\mathcal{P}$. Let $t \leftarrow t + t_i'$.
15: **end for**
16: Output the partition $\mathcal{P} = \{P_1, P_2, \cdots, P_t\}$ and the centers $c : \mathcal{P} \to P$.

---

At a high level, the algorithm iteratively processes $R = 1, 2, 4, \ldots, \Delta$ (recall that in the case of our input, the distance of each pair of points is between $1$ and $O(\sqrt{d})$). For each value of $R$, the algorithm needs to access a $C$-approximate $(R, k)$-near neighbor graph $G$, and then compute a $\beta$-ruling set of a subgraph of $G^2$. The algorithm maintains the following invariants at the end of the iteration with respect to the value of $R$:

  (a) Every point $p$ satisfies $\rho_k(p) \leq R$ must be assigned to some cluster.
  (b) The radius of each cluster is at most $O(C \cdot R)$
  (c) The size of each cluster is at least $k$.

For the correctness of Algorithm 2 and more details, we refer the readers to Section 3.3 in [EMMZ22].

Combining with Lemma B.3 and Lemma B.4, we can prove the correctness of our Theorem 1.2.

