# OpenReview forum: "Nearly-Linear Time and Massively Parallel Algorithms for $k$-anonymity"
_NeurIPS.cc/2025/Conference — NeurIPS 2025 poster_

### Official Review · Reviewer_T5kh · 2025-07-02

**Clarity:** 2
**Significance:** 3
**Originality:** 2
**Rating:** 4
**Confidence:** 3

**Summary:**

This paper studies the $k$-anonymity problem, which is the task of partitioning a dataset into groups of at least $k$ records each, such that every individual becomes indistinguishable from at least $k-1$ others on quasi-identifiers. The authors propose a new $O(k)$ approximation algorithm with near-linear running time and extend this algorithm to the Massively Parallel Computation model. At a technical level, the author notice that the $O(k)$ approximation for $k$-anonymity can be reduced to the minimum-size constrained clustering problem and apply the r-gather algorithm with pointwise guarantee in [EMMZ22]. They also give a hardness result on the following single-point $k$-anonymity problem, where the goal is to select $k - 1$ additional records to make a given record indistinguishable while minimizing the number of hidden attributes among these $k$ points.

**Questions:**

**Questions:**

* It would be helpful if the authors could include an explicit analysis of the space complexity. The current algorithm seems to incur an extra $n^{o(1)}$ factor due to the use of LSH and this leads to an extra $n^{o(1)}$ factor in the MPC setting. In addition, it would be valuable to include a comparison with the $O(n^2k)$ space complexity of the algorithm in [AFK+05].
* Could you clarify the main technical contribution of your result? As mentioned above, the main results and ideas, especially in the MPC part, seem largely drawn from [EMMZ22].
* Could you include a comparison of the running time in the experimental section?
* In theorem 1.2, how the total space depends on $\epsilon$?

**Ethical Concerns:**

["NO or VERY MINOR ethics concerns only"]

**Final Justification:**

The reason is given in my previous comment.

**Limitations:**

Yes

**Paper Formatting Concerns:**

No major formatting issues found in the paper.

**Quality:**

2

**Strengths And Weaknesses:**

Strengths:

* The result improves the sequential algorithm in [EMMZ22], resulting in a better overall running time of
  $\tilde O(nd + n·(n/k)^{1/C² + o(1)})$, which is better than $O(n^2k)$ in [AFK+05]. Furthermore, an MPC algorithm for this problem is provided.
* A lower bound for a related problem, i.e., single-point k-anonymity problem, is given.

Weaknesses:

*  The main weakness in my opinion is the lack of novelty. The algorithms mainly based on a direct implementation of [EMMZ22]. The MPC part also appears to be a straightforward application of the framework from [EMMZ22], without a clear connection to the sequential improvements presented earlier.
* Though a lower bound for the single-point $k$-anonymity is given, but its connection to the (lower bound of) the original problem is unclear.
* While the paper’s main contribution lies in improving the time complexity, the experiments only focus on comparing the number of hidden attributes, with no evaluation of runtime performance.


Typos:

* On pages 6–7, the use of $O(1)$ seems inaccurate and should likely be $O(C)$.

---

> ### Author Rebuttal · Authors · 2025-07-30
>
> We thank the reviewer for the detailed and constructive feedback. Below we address the specific questions.
>
> ---
>
> **Space complexity**:
>
> We thank the reviewer for pointing this out. We will include a more detailed space complexity analysis in the next version of the paper. It is correct that the use of LSH introduces additional space overhead. However, we note that in our algorithm, LSH is always computed over a subset of size $|J'| = (n \log n) / k$ (though multiple times), rather than over the entire dataset. As a result, the overall space complexity of our algorithm is:
>
> $$
> O\left(nd + n \cdot \left(\frac{n}{k}\right)^{1/C^2 + o(1)}\right)
> $$
>
> In comparison, the algorithm in [AFK+05] does not provide an explicit space complexity discussion, but to the best of our knowledge, its space usage is $O(nd)$.
>
> In Theorem 1.2, the total space complexity does not depend on $\epsilon$, as $\epsilon$ serves as a trade-off parameter between accuracy and parallel runtime.
>
> ---
>
> **Technical contributions**:
>
> - We  present a reduction showing that achieving an $O(k)$-approximation for $k$-anonymity can be reduced to solving the *minimum-size constrained clustering* problem with an $O(1)$ pointwise approximation guarantee under squared $\ell_2$ distance . To our knowledge, this reduction is new to the literature.
>
> - Our algorithm for solving the minimum-size constrained clustering problem is inspired by [EMMZ22], where locality-sensitive hashing (LSH) is used. However, as noted in lines 179–184, directly adapting their algorithm leads to a runtime of $n^{1 + 1/C^2 + o(1)} k$, which can be as large as $n^2$ when $k = \Theta(n)$. To overcome this, we incorporate a random sampling strategy that reduces $k$-nearest neighbor computations to 1-nearest neighbor computations, which to our knowledge, is generic and could have independent applications.
>
> - Finally, we provide a lower bound result for *single-point* $k$-anonymity (a local version of the original problem), showing that computing an $o(k)$-approximation is hard. Extending this result to a reduction from the global version remains an open and interesting direction for future work.
>
> ---
>
> **Runtime in experiments**:
>
> We thank the reviewer for raising this important point. We conducted additional experiments comparing our method with the most recent heuristic algorithm proposed in [ZWL+18]. We note that this algorithm performs on real-world data, but also has high computational complexity.The core idea of this algorithm involves:
> 1. Iteratively selecting a new cluster center as the point with the highest average distance from existing centers, and
> 2. Assigning the $k - 1$ nearest neighbors of that point to the same cluster.
>
> This strategy, while intuitive, results in at least $\Omega(n^2)$ time complexity, even when we disregard other parameters such as $d$ and $k$.
>
> To empirically assess runtime performance, we conducted experiments comparing the following algorithms:
>
> - **Mondrian**: We used the publicly available Python implementation (referenced on page 12 of [EEMM24]).
>
> - **Our algorithm**: Implemented in C++ for efficiency.
>
> - **[ZWL+18]**: Since no official implementation was available, we implemented the algorithm ourselves in C++. We applied the same optimization strategies as in our own implementation, including multi-threading where applicable and pre-processing steps to speed up pairwise distance computations.
>
> The experimental results confirm our theoretical expectations. On the *Adult* dataset with $k = 10$:
> - **Mondrian** completed in approximately **1 second**,
> - **Our algorithm** completed in **3.78 seconds**, and
> - **[ZWL+18]** completed in **689.15 seconds** (here the run time increases a lot as $k$ decreases, see the below table for details).
>
>
> Moreover, motivated by the performance bottleneck of [ZWL+18], we investigated whether the core heuristic could be accelerated using ideas from our own algorithm. We found that substantial speed-ups are indeed possible, with minimal impact on performance.
>
> In particular, we introduce the following modifications:
>
> 1. For the center selection step (Step 1), instead of computing the average distance for all points, we randomly sample a fixed number of candidate points.
>
> 2. Once a new center is chosen, we leverage Locality-Sensitive Hashing (LSH) to efficiently compute its approximate $k$-nearest neighbors. This process is similar to Lemmas 3.5 and 3.6 in our submission.
>
> In our experiments, these modifications lead to a substantial reduction in runtime while preserving performance comparable to the original heuristic. Notably, the improved variant consistently outperforms the results presented in Section 5 of our current submission.
>
> Below, we report the experimental results. We refer to the improved version of our new algorithm as **Ours (Heuristic)**:
>
> | Dataset       | Algorithm         | Runtime (s) | Number of hidden entries                  |
> |---------------|-------------------|-------------|-------------------------------------------|
> | Adult (k = 10)| Mondrian          | ~1.00       | 116214                                    |
> |               | ZWL+18            | 689.15      | 63617                                     |
> |               | Ours (Heuristic)  | 13.82       | 65209                                     |
> | Adult (k = 15)| Mondrian          | ~1.00       | 126422                                    |
> |               | ZWL+18            | 308.95      | 72284                                     |
> |               | Ours (Heuristic)  | 7.35        | 72956                                     |
> | Adult (k = 20)| Mondrian          | ~1.00       | 133783                                    |
> |               | ZWL+18            | 176.78      | 72856                                     |
> |               | Ours (Heuristic)  | 4.896       | 78724                                     |
>
> In summary, our algorithm achieves nearly-linear runtime in theory. Empirically, we also demonstrate that our algorithmic ideas can be adapted to existing heuristic methods, leading to significant speed-ups while preserving comparable performance.

---

> > ### Comment · Reviewer_T5kh · 2025-08-07
> >
> > Thank you for your response. I will raise the score from 3 to 4. However, I'm still not convinced about the relationship between the upper bound and lower bound results.

---

### Official Review · Reviewer_tmta · 2025-07-02

**Clarity:** 2
**Significance:** 3
**Originality:** 3
**Rating:** 4
**Confidence:** 4

**Summary:**

The authors develop approximation guarantees for k-anonymity, with one of the main goals being to provide an O(k) approximation algorithm with running time linear in the size of the input. They also develop an approximation algorithm in the MPC model as well as a lower bound on a related problem (whose connection with k-anonymity is still unclear). Their theoretical results are complemented with an experimental evaluation on some real-world datasets.

**Questions:**

See my comments above

**Ethical Concerns:**

["NO or VERY MINOR ethics concerns only"]

**Final Justification:**

I still believe the results are interesting however I also agree with the issues pointed out by Reviewer T5kh.

**Limitations:**

yes

**Quality:**

3

**Strengths And Weaknesses:**

+ the problem studied in the paper is well motivated and relevant to NeurIPS
+ the paper is fairly well written and organized

- although the most important parts of the paper are clear, there is room for improvement
- the experimental evaluation is limited in scope
- the code is not publicly available, as far as I could check


more detailed comments:
- the experimental evaluation is limited in scope. One way to improve it could be to include an experimental evaluation of their approach against AFK+05 and ideally show that the two approaches have "similar" guarantees while their approach is much faster.
- the code should be made publicly available so as to foster reproducibility

some comments about the presentation:
- line 36 report the full running time [MW04]
- line 48 runs in linear in n time => with running time linear in n
- line 49 probable approximation guarantees => provable approximation guarantees
- theorem 1.1. for every C>1?
- section 3, define the the minimum size constrained clustering problem

---

> ### Author Rebuttal · Authors · 2025-07-30
>
> We thank the reviewer for the detailed and constructive feedback. Below we address the specific concerns.
>
> ---
>
> **Runtime in experiments**:
>
> We thank the reviewer for raising this important point. We conducted additional experiments comparing our method with the most recent heuristic algorithm proposed in [ZWL+18]. We note that this algorithm performs on real-world data, but also has high computational complexity.The core idea of this algorithm involves:
> 1. Iteratively selecting a new cluster center as the point with the highest average distance from existing centers, and
> 2. Assigning the $k - 1$ nearest neighbors of that point to the same cluster.
>
> This strategy, while intuitive, results in at least $\Omega(n^2)$ time complexity, even when we disregard other parameters such as $d$ and $k$.
>
> To empirically assess runtime performance, we conducted experiments comparing the following algorithms:
>
> - **Mondrian**: We used the publicly available Python implementation (referenced on page 12 of [EEMM24]).
>
> - **Our algorithm**: Implemented in C++ for efficiency.
>
> - **[ZWL+18]**: Since no official implementation was available, we implemented the algorithm ourselves in C++. We applied the same optimization strategies as in our own implementation, including multi-threading where applicable and pre-processing steps to speed up pairwise distance computations.
>
> The experimental results confirm our theoretical expectations. On the *Adult* dataset with $k = 10$:
> - **Mondrian** completed in approximately **1 second**,
> - **Our algorithm** completed in **3.78 seconds**, and
> - **[ZWL+18]** completed in **689.15 seconds** (here the run time increases a lot as $k$ decreases, see the below table for details).
>
>
> Moreover, motivated by the performance bottleneck of [ZWL+18], we investigated whether the core heuristic could be accelerated using ideas from our own algorithm. We found that substantial speed-ups are indeed possible, with minimal impact on performance.
>
> In particular, we introduce the following modifications:
>
> 1. For the center selection step (Step 1), instead of computing the average distance for all points, we randomly sample a fixed number of candidate points.
>
> 2. Once a new center is chosen, we leverage Locality-Sensitive Hashing (LSH) to efficiently compute its approximate $k$-nearest neighbors. This process is similar to Lemmas 3.5 and 3.6 in our submission.
>
> In our experiments, these modifications lead to a substantial reduction in runtime while preserving performance comparable to the original heuristic. Notably, the improved variant consistently outperforms the results presented in Section 5 of our current submission.
>
> Below, we report the experimental results. We refer to the improved version of our new algorithm as **Ours (Heuristic)**:
>
> | Dataset       | Algorithm         | Runtime (s) | Number of hidden entries                  |
> |---------------|-------------------|-------------|-------------------------------------------|
> | Adult (k = 10)| Mondrian          | ~1.00       | 116214                                    |
> |               | ZWL+18            | 689.15      | 63617                                     |
> |               | Ours (Heuristic)  | 13.82       | 65209                                     |
> | Adult (k = 15)| Mondrian          | ~1.00       | 126422                                    |
> |               | ZWL+18            | 308.95      | 72284                                     |
> |               | Ours (Heuristic)  | 7.35        | 72956                                     |
> | Adult (k = 20)| Mondrian          | ~1.00       | 133783                                    |
> |               | ZWL+18            | 176.78      | 72856                                     |
> |               | Ours (Heuristic)  | 4.896       | 78724                                     |
>
> In summary, our algorithm achieves nearly-linear runtime in theory. Empirically, we also demonstrate that our algorithmic ideas can be adapted to existing heuristic methods, leading to significant speed-ups while preserving comparable performance.
>
> ---
>
> Code: As the rebuttal policy does not allow uploading supplementary material or external links, we will release our code publicly with the next version of the paper.

---

### Official Review · Reviewer_ozzy · 2025-07-02

**Clarity:** 3
**Significance:** 2
**Originality:** 3
**Rating:** 4
**Confidence:** 3

**Summary:**

This paper studies the natural optimization version of $k$-anonymity: given a database, what are minimum number of cell suppressions needed in order to make the database satisfy $k$-anonymity?  Previous papers give $O(k)$-approximations with essentially quadratic runtime, or give better $O(\log k)$-approximations but with extremely large $O(n^{2k})$ runtime.  This paper gives an $O(k)$-approximation that takes nearly linear time, and also gets it to work in the parallel/distributed MPC model using polylogarithmic rounds and an arbitrarily small polynomial amount of extra space.  They also prove that, under a (somewhat nonstandard but reasonable) complexity assumption, if we require our algorithm to have running time that is polynomial in $k$, then we can't get approximations that are significantly better than $\Omega(k)$.

The main technique is to first translate the $k$-anonymity problem into a "minimum size constrained clustering" problem in the $\ell_2^2$ distance metric.  This problem has been studied in the MPC model before, but that algorithm does not directly give a fast enough algorithm here.  So the authors show how to cleverly use LSH and random sampling to speed this up.

**Questions:**

- Can you provide some justification for why this problem is important in an era when DP has become the gold standard for privacy?
- How does your algorithm perform with respect to running time experimentally?

**Ethical Concerns:**

["NO or VERY MINOR ethics concerns only"]

**Final Justification:**

The authors ran some new experiments which seem good, and we have had a discussion about the importance of k-anonymity.  I am still quite skeptical, for the reasons in my rebuttal and responses, but the authors found a few more citations to back them up.  So I've raised my score to 4.  I think this would be a reasonable NeurIPS paper, but I certainly wouldn't push for acceptance.

**Limitations:**

yes

**Quality:**

2

**Strengths And Weaknesses:**

Strengths:
  - This algorithms seems reasonably practical, and getting a quadratic running time down to near-linear is certainly an important goal and achievement.
  - The techniques seem to be nontrivial and above the technical bar.

Weaknesses:
  - My main concern is the motivation for this problem.  $k$-anonymity was an important concept at one point in time, but as far as I can tell has been completely superseded by differential privacy.  This can be seem in the references in this paper, which generally are >10 years old.  As far as I can tell, the only discussion about this is at the end of Section 1.2, where they say "Recently, the work of [EEMM24] studied the smooth k-anonymity problem on a binary dataset, where they provide a detailed discussion comparing k-anonymity and differential privacy. Their findings suggest that in certain scenarios, k-anonymity can be more suitable than differential privacy."  To me, that does not seem like sufficient justification.  What are those scenarios?  Do they correspond to settings where we want to get $k$-anonymity by suppression?  I have no idea, since that discussion is not included in this paper.  They just outsource it to EEMM24.
  - The experiments are a bit weak.  They show that the theoretical algorithm does not perform particularly well, but a simple variant of it does.  But the main point of this paper is about running time, and I don't see any experiments showing that this new algorithm is actually *faster* than previous algorithms.  Moreover, they do not compare to other heuristics because "they have runtime at least $O(n^2)$".  But, of course, a worst-case running time of $O(n^2)$ does not necessarily mean that it will actually be slower in practice.  Since the point of this paper is to give a faster algorithm, I would have expected experiments showing that their algorithm is faster than the previous quadratic time algorithm on realistic datasets, and moreover is at least as fast as previous heuristic approaches on those datasets.

---

> ### Author Rebuttal · Authors · 2025-07-30
>
> We thank the reviewer for the detailed and constructive feedback. Below we address the specific concerns.
>
> ---
>
> **Motivation for k-anonymity**:
>
> We appreciate the question regarding our motivation for using *k*-anonymity over differential privacy (DP), and we will provide additional explanation in the revised version of the manuscript.
>
> In [EEMM 24], the authors formally prove the following:
>
> > Let $\mathcal{M}$ be an arbitrary mechanism that satisfies $\epsilon$-edge differential privacy. Then, in order to achieve $E[J(\mathcal{M}(G), G)] \ge \alpha$, it must hold that $\epsilon = \Omega(\log (\alpha^2 n m))$, where $J(\cdot, \cdot)$ denotes the Jaccard similarity.
>
> This result suggests that achieving high utility (i.e., preserving the structure of the original graph) under differential privacy requires a large value of $\epsilon$. However, when $\epsilon$ is large, the algorithm likely maintains the graph G unmodified thus exposing users to re-identification risks. Recall that the guarantee provided by $\epsilon$-DP is: $\Pr[\mathcal{M}(G) \in A] \le e^{\epsilon} \Pr[\mathcal{M}(G') \in A],$ which becomes nearly vacuous for large $\epsilon$.
>
> In contrast, *k*-anonymity offers a different privacy-utility tradeoff. Prior work shows that if the optimal anonymized graph $E_{opt}$ satisfies $J(E, E_{opt}) \ge 1 - O(1/\log k)$, then efficient algorithms can find solutions with comparable utility. Furthermore, in the case of *smooth* k-anonymity, where edge additions and deletions are allowed, the required assumption can be relaxed to $J(E, E_{opt}) \ge O(1)$.
>
> In recent years, several notable works on data anonymity have been published at top machine learning venues. Examples include:
>
> - Hossein Esfandiari et al., *Anonymous Bandits for Multi-User Systems* (NeurIPS, 2022)
> - Alessandro Epasto et al., *Smooth Anonymity for Sparse Graphs* (WWW, 2024)
> - Xinyi Zheng et al., *Building K-Anonymous User Cohorts with Consecutive Consistent Weighted Sampling (CCWS)* (SIGIR, 2023)
> - Krzysztof Choromanski et al., *Adaptive Anonymity via b-Matching* (NeurIPS, 2013)
>
> These works highlight interest in anonymity and privacy-preserving algorithms in machine learning and other communities. Beyond foundational and algorithmic work, we observed  $k$-anonymity has also been applied in several domains recently. Examples include:
>
> -  Edvinas Kruminis et al., *BB-FLoC: A Blockchain-based Targeted Advertisement Scheme with K-Anonymity* (Distributed Ledger Technologies: Research and Practice, 2024)
> -  Zhaowei Hu et al., *KAIM: A distributed K-anonymity selection mechanism based on dual incentives* (Ad Hoc Networks Journal. 2025)
> - Stylianos Karagiannis et al., *Mastering data privacy: leveraging K-anonymity for robust health data sharing* (International Journal of Information Security, 2024)
> ---
> ---
>
> **Runtime in experiments**:
>
> We thank the reviewer for raising this important point. We note that the algorithms we mention indeed have a slow runtime on real-world data. For instance, the most recent heuristic algorithm proposed in [ZWL+18] exhibits high computational complexity. The core idea of this algorithm involves:
> 1. Iteratively selecting a new cluster center as the point with the highest average distance from existing centers, and
> 2. Assigning the $k - 1$ nearest neighbors of that point to the same cluster.
>
> This strategy, while intuitive, results in at least $\Omega(n^2)$ time complexity, even when we disregard other parameters such as $d$ and $k$.
>
> To empirically assess runtime performance, we conducted experiments comparing the following algorithms:
>
> - **Mondrian**: We used the publicly available Python implementation (referenced on page 12 of [EEMM24]).
>
> - **Our algorithm**: Implemented in C++ for efficiency.
>
> - **[ZWL+18]**: Since no official implementation was available, we implemented the algorithm ourselves in C++. We applied the same optimization strategies as in our own implementation, including multi-threading where applicable and pre-processing steps to speed up pairwise distance computations.
>
> The experimental results confirm our theoretical expectations. On the *Adult* dataset with $k = 10$:
> - **Mondrian** completed in approximately **1 second**,
> - **Our algorithm** completed in **3.78 seconds**, and
> - **[ZWL+18]** completed in **689.15 seconds** (here the run time increases a lot as $k$ decreases, see the below table for details).
>
>
> Moreover, motivated by the performance bottleneck of [ZWL+18], we investigated whether the core heuristic could be accelerated using ideas from our own algorithm. We found that substantial speed-ups are indeed possible, with minimal impact on performance.
>
> In particular, we introduce the following modifications:
>
> 1. For the center selection step (Step 1), instead of computing the average distance for all points, we randomly sample a fixed number of candidate points.
>
> 2. Once a new center is chosen, we leverage Locality-Sensitive Hashing (LSH) to efficiently compute its approximate $k$-nearest neighbors. This process is similar to Lemmas 3.5 and 3.6 in our submission.
>
> In our experiments, these modifications lead to a substantial reduction in runtime while preserving performance comparable to the original heuristic. Notably, the improved variant consistently outperforms the results presented in Section 5 of our current submission.
>
> Below, we report the experimental results. We refer to the improved version of our new algorithm as **Ours (Heuristic)**:
>
> | Dataset       | Algorithm         | Runtime (s) | Number of hidden entries                  |
> |---------------|-------------------|-------------|-------------------------------------------|
> | Adult (k = 10)| Mondrian          | ~1.00       | 116214                                    |
> |               | ZWL+18            | 689.15      | 63617                                     |
> |               | Ours (Heuristic)  | 13.82       | 65209                                     |
> | Adult (k = 15)| Mondrian          | ~1.00       | 126422                                    |
> |               | ZWL+18            | 308.95      | 72284                                     |
> |               | Ours (Heuristic)  | 7.35        | 72956                                     |
> | Adult (k = 20)| Mondrian          | ~1.00       | 133783                                    |
> |               | ZWL+18            | 176.78      | 72856                                     |
> |               | Ours (Heuristic)  | 4.896       | 78724                                     |
>
> In summary, our algorithm achieves nearly-linear runtime in theory. Empirically, we also demonstrate that our algorithmic ideas can be adapted to existing heuristic methods, leading to significant speed-ups while preserving comparable performance.

---

> > ### Comment · Reviewer_ozzy · 2025-08-04
> >
> > Thank you for the response.  I'm still not particularly convinced by k-anonymity here.  Since the theorem from [EEMM24] is about Jaccard similarity, it seems to basically say that under edge-DP you need to tailor your synthetic graph to the actual problem you are trying to solve (like all previous work on edge-DP does) -- you can't output a synthetic graph and expect it to be close for all problems (like you'd get if they're close under Jaccard).  But this feels basically fundamental to privacy here, and actually makes me very suspicious of k-anonymity.  If better utility is possible for k-anonymity, then it feels to me not like justification for k-anonymity, but rather evidence that k-anonymity is too weak a privacy notion (which was some of the original motivation for DP).
> >
> > So since I still think the problem lacks motivation, I'm going to leave my score where it is.

---

> > > ### Author Response · Authors · 2025-08-05
> > >
> > > We appreciate the reviewer's thoughtful feedback. We agree that Differential Privacy provides the strongest theoretical privacy guarantees. However, we believe that the landscape of data privacy is not a simple hierarchy with DP at the top, but rather a space of trade-offs where different tools are suited for different practical needs. Our motivation for studying *k*-anonymity is grounded in its practical relevance and observed use in various real-world applications.
> > >
> > >
> > > - **Practicality and Scalability in Real-World Systems**: In particular, achieving the strongest level of privacy is not the only consideration in real-world systems. Practical deployments must also account for constraints such as efficiency, scalability, and data utility. These constraints are especially prominent in large-scale applications, where mechanisms like k-anonymity often offer a more favorable trade-off. It is our understanding that leading organizations such as Google have explored and deployed *k*-anonymity-based approaches for these reasons. For instance, as discussed in [2], Google’s **Federated Learning of Cohorts (FLoC)** project, which is part of the Chrome Privacy Sandbox, replaces individually identifying third-party cookies with large anonymous cohorts of users. The underlying algorithm clusters millions of browser users into cohorts that each satisfy a minimum-size constraint, effectively instantiating a *k*-anonymity problem. Because both the number of users and the number of cohorts can be very large, such settings demand high efficiency. We therefore view *k*-anonymity as offering a favorable trade-off between privacy, utility, and computational efficiency in large-scale log-style data. Although public documentation of such deployments is limited, ongoing research efforts on related topics (e.g., [1–4]) with authors from major industrial labs (e.g., Google Research) suggest that *k*-anonymity remains an area of active interest in both academia and industry.
> > >
> > > - **Verifiability and Trust**: We would like to emphasize that k-anonymity is a property of the **output dataset**, whereas DP is a property of the **algorithm** that generated it. This means a k-anonymized dataset can be independently audited and its privacy property can be empirically verified by a third party. This aligns with the principle of accountability sought by regulators (e.g., as discussed in the executive summary in EU's Article 29 WP216 opinion on anonymization, where the document also discusses the pros and cons for each method). In contrast, verifying a DP guarantee requires trusting that the generating algorithm was implemented and executed correctly, which can be a much more difficult proposition.
> > >
> > > - **The Practical Trade-off between $k$ and $\epsilon$**: while the original DP framework proposed using small $\epsilon$ values (e.g., $\epsilon < 1$), real-world deployments often rely on significantly larger values, which can weaken the intended privacy guarantee. As a result, some practitioner might find the concrete and interpretable nature of *k*-anonymity levels (e.g., *k* = 10) more intuitive and actionable than that of a high $\epsilon$. A recent position paper further explores these practical considerations for DP ([5]).
> > >
> > >
> > > **References**
> > >
> > > 1. Alessandro Epasto et al., *Smooth Anonymity for Sparse Graphs*
> > > 2. Alessandro Epasto et al., *Massively Parallel and Dynamic Algorithms for Minimum Size Clustering*
> > > 3. CJ Carey et al., *Measuring Re-identification Risk*
> > > 4. Alina Ene et al., *Maximum Coverage in Turnstile Streams with Applications to Fingerprinting Measures*
> > > 5. Kareem Amin et al., *Practical Considerations for Differential Privacy*

---

> > > > ### Comment · Reviewer_ozzy · 2025-08-05
> > > >
> > > > These are fair points.  I've been convinced to raise my score to 4 (from 3).  I'm still concerned that this discussion is not actually in the paper, though, and I strongly suggest that the authors revise to include a discussion defending k-anonymity.
> > > >
> > > > For me personally, in order to be really convinced, I would need to see not just "here are other papers that study k-anonymity", but rather "here's at least one paper that has an actual discussion of why k-anonymity might still be important today".  As far as I can tell that is not actually in any of the cited papers, and as far as I know might not exist.  I somewhat suspect it does not exist because I am still super skeptical that k-anonymity is actually important, which is why I'm still only giving a 4.  There's no question that it was used in privacy sandbox at Google, but my understanding is that privacy sandbox is basically dead (see, e.g., https://digiday.com/marketing/google-explores-possibilities-for-the-privacy-sandbox-after-cookie-u-turn/).  And even if privacy sandbox *were* still being developed, "Google says that this notion of privacy allows them to make money while still providing enough privacy" is not actually a particularly convincing argument to me -- of course every industrial company is going to try to claim that weak notions of privacy are actually strong.  The real question is whether there are any actual privacy advocates who still believe in k-anonymity at this point.  My guess is that there aren't but I'm not totally sure of that.

---

### Official Review · Reviewer_Urpi · 2025-07-03

**Clarity:** 3
**Significance:** 3
**Originality:** 3
**Rating:** 6
**Confidence:** 4

**Summary:**

Give the first $\mathbf{O}(\mathbf{k})$-approximation in nearly-linear time
$$
\widetilde{O}\left(n d+n(n / k)^{1 / C^2+o(1)}\right)
$$
matching the best known approximation ratio while improving runtime over  $\mathrm{O}\left(\mathrm{n}^2 \mathrm{k}\right)$ algorithms previously.

**Questions:**

1.) What are the constant factors like in greater detail?

2.) How do you set the JL target dimension ?

3.) What are the specific setups where k-anonymity is better than differential privacy?

**Ethical Concerns:**

["NO or VERY MINOR ethics concerns only"]

**Final Justification:**

I uphold my score at 6 and support this paper based on the fact that k-anonymity although not the best method, is still used in practice and is being studied by researchers and the authors have moved the needle on this front.

**Limitations:**

Dependence on the hypergraph dense Vs. random conjecture,

**Quality:**

4

**Strengths And Weaknesses:**

Provides a scalable MPC version and prove hardness of single-point k-anonymity problem under a Dense vs. Random conjecture, ruling out k^{1–o(1/C)}-approximation in polynomial time.

The O(k)-approximation with nearly-linear runtime is a significant theoretical advance over prior O(n²k) methods.

The paper is quite well written.

Minor: The authors mention: "Recently, the work of [EEMM24] studied the smooth $k$-anonymity problem on a binary dataset, where they provide a detailed discussion comparing $k$-anonymity and differential privacy. Their findings suggest that in certain scenarios. $k$-anonvmity can be more suitable than differential privacv (DP)". That said, in this PNAS paper they have proved that k-anonymity unlike DP does not satisfy the GDPR clause of preventing identifiability: https://www.pnas.org/doi/10.1073/pnas.1914598117 . Thereby it would be good to explain the cases/setups where k-anonymity is better than DP. That helps better motivate the paper as k-anonymity is an older technique.

---

> ### Author Rebuttal · Authors · 2025-07-30
>
> We thank the reviewer for the detailed and constructive feedback. Below we address the specific questions.
>
> ---
>
> Motivation for k-anonymity:
>
> We appreciate the question regarding our motivation for using *k*-anonymity over differential privacy (DP), and we will provide additional explanation in the revised version of the manuscript.
>
> In [EEMM 24], the authors formally prove the following:
>
> > Let $\mathcal{M}$ be an arbitrary mechanism that satisfies $\epsilon$-edge differential privacy. Then, in order to achieve $E[J(\mathcal{M}(G), G)] \ge \alpha$, it must hold that $\epsilon = \Omega(\log (\alpha^2 n m))$, where $J(\cdot, \cdot)$ denotes the Jaccard similarity.
>
> This result suggests that achieving high utility (i.e., preserving the structure of the original graph) under differential privacy requires a large value of $\epsilon$. However, when $\epsilon$ is large, the algorithm likely maintains the graph G unmodified thus exposing users to re-identification risks. Recall that the guarantee provided by $\epsilon$-DP is: $\Pr[\mathcal{M}(G) \in A] \le e^{\epsilon} \Pr[\mathcal{M}(G') \in A],$ which becomes nearly vacuous for large $\epsilon$.
>
> In contrast, *k*-anonymity offers a different privacy-utility tradeoff. Prior work shows that if the optimal anonymized graph $E_{opt}$ satisfies $J(E, E_{opt}) \ge 1 - O(1/\log k)$, then efficient algorithms can find solutions with comparable utility. Furthermore, in the case of *smooth* k-anonymity, where edge additions and deletions are allowed, the required assumption can be relaxed to $J(E, E_{opt}) \ge O(1)$.
>
> In recent years, several notable works on data anonymity have been published at top machine learning venues. Examples include:
>
> - Hossein Esfandiari et al., *Anonymous Bandits for Multi-User Systems* (NeurIPS, 2022)
> - Alessandro Epasto et al., *Smooth Anonymity for Sparse Graphs* (WWW, 2024)
> - Xinyi Zheng et al., *Building K-Anonymous User Cohorts with Consecutive Consistent Weighted Sampling (CCWS)* (SIGIR, 2023)
> - Krzysztof Choromanski et al., *Adaptive Anonymity via b-Matching* (NeurIPS, 2013)
>
> These works highlight interest in anonymity and privacy-preserving algorithms in machine learning and other communities.
> Beyond foundational and algorithmic work, we observed $k$-anonymity has also been applied in several domains recently. Examples include:
>
> -  Edvinas Kruminis et al., *BB-FLoC: A Blockchain-based Targeted Advertisement Scheme with K-Anonymity* (Distributed Ledger Technologies: Research and Practice, 2024)
> -  Zhaowei Hu et al., *KAIM: A distributed K-anonymity selection mechanism based on dual incentives* (Ad Hoc Networks Journal. 2025)
> - Stylianos Karagiannis et al., *Mastering data privacy: leveraging K-anonymity for robust health data sharing* (International Journal of Information Security, 2024)
> ---
>
> Constant factors:
>
> We agree with the reviewer that our theoretical bounds include several constant factors. However, our goal in this work is not to optimize these constants. Specifically:
>
> - In Lemma 3.1, a factor of 8 appears;
> - In the clustering step, the use of the triangle inequality introduces a factor of 2;
> - Finally, the LSH-based algorithm introduces an additional constant $c_u$.
>
> ---
>
> Johnson-Lindenstrauss (JL) target dimension:
>
> Here we need to preserve all pairwise distances within a $(1 \pm 0.1)$ multiplicative error, a JL projection to $O(\log n)$ dimensions suffices.
>
> ---
>
> More Experiments:
>
> Motivated by the reviewers’ feedback, we conducted additional experiments comparing our method with the most recent heuristic algorithm proposed in [ZWL+18]. We note that this algorithm performs on real-world data, but also has high computational complexity. The core idea of this algorithm involves:
> 1. Iteratively selecting a new cluster center as the point with the highest average distance from existing centers, and
> 2. Assigning the $k - 1$ nearest neighbors of that point to the same cluster.
>
> This strategy, while intuitive, results in at least $\Omega(n^2)$ time complexity, even when we disregard other parameters such as $d$ and $k$.
>
> To empirically assess runtime performance, we conducted experiments comparing the following algorithms:
>
> - **Mondrian**: We used the publicly available Python implementation (referenced on page 12 of [EEMM24]).
>
> - **Our algorithm**: Implemented in C++ for efficiency.
>
> - **[ZWL+18]**: Since no official implementation was available, we implemented the algorithm ourselves in C++. We applied the same optimization strategies as in our own implementation, including multi-threading where applicable and pre-processing steps to speed up pairwise distance computations.
>
> The experimental results confirm our theoretical expectations. On the *Adult* dataset with $k = 10$:
> - **Mondrian** completed in approximately **1 second**,
> - **Our algorithm** completed in **3.78 seconds**, and
> - **[ZWL+18]** completed in **689.15 seconds** (here the run time increases a lot as $k$ decreases, see the below table for details).
>
>
> Moreover, motivated by the performance bottleneck of [ZWL+18], we investigated whether the core heuristic could be accelerated using ideas from our own algorithm. We found that substantial speed-ups are indeed possible, with minimal impact on performance.
>
> In particular, we introduce the following modifications:
>
> 1. For the center selection step (Step 1), instead of computing the average distance for all points, we randomly sample a fixed number of candidate points.
>
> 2. Once a new center is chosen, we leverage Locality-Sensitive Hashing (LSH) to efficiently compute its approximate $k$-nearest neighbors. This process is similar to Lemmas 3.5 and 3.6 in our submission.
>
> In our experiments, these modifications lead to a substantial reduction in runtime while preserving performance comparable to the original heuristic. Notably, the improved variant consistently outperforms the results presented in Section 5 of our current submission.
>
> Below, we report the experimental results. We refer to the improved version of our new algorithm as **Ours (Heuristic)**:
>
> | Dataset       | Algorithm         | Runtime (s) | Number of hidden entries                  |
> |---------------|-------------------|-------------|-------------------------------------------|
> | Adult (k = 10)| Mondrian          | ~1.00       | 116214                                    |
> |               | ZWL+18            | 689.15      | 63617                                     |
> |               | Ours (Heuristic)  | 13.82       | 65209                                     |
> | Adult (k = 15)| Mondrian          | ~1.00       | 126422                                    |
> |               | ZWL+18            | 308.95      | 72284                                     |
> |               | Ours (Heuristic)  | 7.35        | 72956                                     |
> | Adult (k = 20)| Mondrian          | ~1.00       | 133783                                    |
> |               | ZWL+18            | 176.78      | 72856                                     |
> |               | Ours (Heuristic)  | 4.896       | 78724                                     |
>
> In summary, our algorithm achieves nearly-linear runtime in theory. Empirically, we also demonstrate that our algorithmic ideas can be adapted to existing heuristic methods, leading to significant speed-ups while preserving comparable performance.

---

> ### Comment · Reviewer_Urpi · 2025-08-02
> **Satisfactory**
>
> The authors gave a satisfactory rebuttal, and I uphold my positive rating of this paper. I believe the paper has to be considered on it's own merits as the solution to a scientific problem (regardless of whether k-anonymity is a old or new method). The proof and algorithmic techniques are still of value. Improvements to the understanding of a classic method like linear regression for instance, still hold merit and further scientific understanding in the current day, for instance.

---

### Note · Authors · 2025-08-14

We sincerely thank all reviewers for their detailed feedback, which has helped us significantly strengthen our paper. We are particularly grateful that, following our discussions, multiple reviewers raised their scores in support of our paper.

A key point of discussion was the motivation for studying $k$-anonymity in the era of differential privacy (DP). As we clarified for Reviewer ozzy, our work is motivated by the practical trade-offs between privacy, utility, and computational efficiency required by large-scale systems. Unlike DP, which is a property of an algorithm, $k$-anonymity is a **verifiable property of the output data**, meaning it can be independently audited. Its continued relevance is also suggested by its presence in some industrial applications and a number of recent publications at top venues. We will add this discussion in the revised version.

Regarding our technical novelty, we thank Reviewer T5kh for pushing us to clarify our contributions beyond prior work like [EMMZ22]. Our main innovations are:

- A **new reduction** from $O(k)$-approximate $k$-anonymity to minimum-size constrained clustering with an $O(1)$ pointwise guarantee.
- A **novel algorithmic speedup** that uses random sampling to reduce $k$-nearest neighbor computations to 1-nearest neighbor, which is crucial for reducing the run time from $n^{1 + 1/C^2 + o(1)} \cdot k$ to $n \cdot (n/k)^{1/C^2 + o(1)}$.

These contributions are not direct applications of [EMMZ22] but rather new techniques required to achieve a nearly-linear time algorithm.

Finally, in response to feedback on our experiments (Reviewers ozzy, tmta), we conducted **new runtime comparisons** against a more recent, computationally-intensive heuristic [ZWL+18]. The experimental results demonstrate that our algorithmic ideas can also be adapted to existing heuristic methods, leading to significant speed-ups while preserving comparable performance. We will add this part in the revised version.


In summary, this paper provides the first nearly-linear time algorithm for k-anonymity with a provable guarantee, a highly scalable MPC version, and a new hardness result for the related single-point k-anonymity problem. We thank the reviewers for their engagement, which has helped us clarify and articulate the significance of these findings.

---

### Decision · Program_Chairs · 2025-09-17

**Decision:**

Accept (poster)

**Comment:**

This paper studies an optimization version of k-anonymity: what is the minimum number of cell suppressions needed to make a database k-anonymous? The authors provide an O(k)-approximation algorithm that runs in near-linear time, improving on previous methods that were either much slower or less accurate. They also adapt it to the Massively Parallel Computation (MPC) model and prove a conditional lower bound that matches their algorithm's approximation ratio. The results tell a "complete story" and the algorithm seems reasonably practical. All reviewers support the paper.